# Thermal acclimation of tropical coral reef fishes to global heat waves

**Jacob L Johansen[1,2†‡]\*, Lauren E Nadler[2,3,4†§]\*, Adam Habary[2], Alyssa J Bowden[5,6], Jodie Rummer[2,4]**

[1]Hawaii Institute of Marine Biology, University of Hawaii, Kaneohe, United States; [2]ARC Centre of Excellence for Coral Reef Studies, James Cook University, Townsville, Australia; [3]Halmos College of Arts and Sciences, Nova Southeastern University, Dania Beach, United States; [4]College of Science and Engineering, James Cook University, Townsville, Australia; [5]CSIRO, Hobart, Australia; [6]Institute of Marine and Antarctic Studies, University of Tasmania, Hobart, Australia

**\*For correspondence:**
jacob.johansen@hawaii.edu (JLJ);
lauren.e.nadler@gmail.com (LEN)

[†]These authors contributed equally to this work

**Present address:** [‡]Hawaii Institute of Marine Biology, University of Hawaii, Kaneohe, United States; [§]Halmos College of Arts and Sciences, Nova Southeastern University, Dania Beach, United States

**Competing interests:** The authors declare that no competing interests exist.

**Abstract** As climate-driven heat waves become more frequent and intense, there is increasing urgency to understand how thermally sensitive species are responding. Acute heating events lasting days to months may elicit acclimation responses to improve performance and survival. However, the coordination of acclimation responses remains largely unknown for most stenothermal species. We documented the chronology of 18 metabolic and cardiorespiratory changes that occur in the gills, blood, spleen, and muscles when tropical coral reef fishes are thermally stressed (+3.0°C above ambient). Using representative coral reef fishes (*Caesio cuning* and *Cheilodipterus quinquelineatus*) separated by >100 million years of evolution and with stark differences in major life-history characteristics (i.e. lifespan, habitat use, mobility, etc.), we show that exposure duration illicited coordinated responses in 13 tissue and organ systems over 5 weeks. The onset and duration of biomarker responses differed between species, with *C. cuning* – an active, mobile species – initiating acclimation responses to unavoidable thermal stress within the first week of heat exposure; conversely, *C. quinquelineatus* – a sessile, territorial species – exhibited comparatively reduced acclimation responses that were delayed through time. Seven biomarkers, including red muscle citrate synthase and lactate dehydrogenase activities, blood glucose and hemoglobin concentrations, spleen somatic index, and gill lamellar perimeter and width, proved critical in evaluating acclimation progression and completion, as these provided consistent evaluation of thermal responses across species.

## Introduction

While sea surface temperatures (SSTs) are expected to rise by 2.0–4.8°C by the end of the century (*Collins et al., 2013*; *IPCC, 2013*; *Pörtner et al., 2019*), a potentially more pressing development is the increasing frequency and severity of extreme acute heating events worldwide (e.g. *Frölicher et al., 2018*; *Hobday et al., 2016*; *IPCC, 2013*; *Pörtner et al., 2019*; *Wernberg et al., 2013*). Examples include the marine heat waves that occurred in the Mediterranean Sea in 2003, in Western Australia in 2011, and on the Great Barrier Reef (GBR) in 2016 and 2017 (*Ainsworth et al., 2016*; *Garrabou et al., 2009*; *Hughes et al., 2017*). These events cause acute increases of up to 5°C above seasonal average SSTs over the course of days and can last for several weeks (*Garrabou et al., 2009*; *Hughes et al., 2017*). The resulting heat stress typically leads to large-scale coral bleaching on coral reefs, mass mortality of fishes and invertebrates, and reduced commercial fisheries catch (*Garrabou et al., 2009*; *Pearce et al., 2011*). To date, climate change studies in marine systems have focused primarily on long-term projected temperature means rather than the accompanying diurnal and monthly extreme temperatures that already occur today (e.g.

*Grenchik et al., 2013*; *Nay et al., 2015*; *Pörtner et al., 2019*; *Rummer et al., 2014*; *Vasseur et al., 2014*). Yet, climate change winners and losers will ultimately be determined by the capacity of individuals and species to compensate for thermal stress in both the short (days, weeks, months) and longer term (years, decades, centuries).

As oceans warm beyond the temperatures under which species have evolved, marine ectotherms will either need to relocate (e.g. to cooler waters or greater depths; *Feary et al., 2014*; *Habary et al., 2017*; *Nay et al., 2015*) or acclimate and adapt to maintain performance (*Johansen and Jones, 2011*; *Sandblom et al., 2016*; *Strobel et al., 2012*; *Tirsgaard et al., 2015*). Acclimation and adaptation are particularly important for the many species that cannot relocate due to lack of mobility or specific resource requirements, including many Arctic and tropical coral reef stenotherms (*Feary et al., 2014*; *Matis et al., 2018*). The species will instead have to safeguard fitness and survival via physiological adjustments (*Donelson et al., 2018*; *Munday et al., 2017*).

The capacity for most species to maintain fitness under the rapid incursion of anthropogenic climate change remains uncertain. Given the critical importance of marine resources for human survival, the likely responses of aquatic ectotherms (especially fishes) to climate change is vigorously debated, and several theories have been proposed to predict these physiological and ecological responses. The Gill-Oxygen Limitation Theory (GOLT) proposes that body size and function in fish is limited by the gills' inability to adjust and supply sufficient oxygen to satisfy increasing metabolic costs under elevated temperatures (*Pauly, 2019*). Similarly, the Oxygen and Capacity Limited Thermal Tolerance (OCLTT) hypothesis proposes cardio-respiratory transport and tissue demand as the main determinants of an organism's performance under ocean warming (*Portner, 2014*). However, mixed empirical evidence has led to a controversy about the exact mechanisms affecting species' performance under elevated temperatures, as neither of the prevailing theories are able to explain all observed responses (reviewed in *Audzijonyte et al., 2019*; *Ern et al., 2017*; *Jutfelt et al., 2014*; *Jutfelt et al., 2018*). More broadly unifying principles are currently lacking (but see *Audzijonyte et al., 2019*; *Clark et al., 2013*; *Ern, 2019*). Recent reviews have, therefore, emphasized the urgent need for cross-disciplinary, mechanistic studies that explore the timescales over which thermal responses occur to assess the processes associated with acclimation and adaptation in thermally sensitive species (*Audzijonyte et al., 2019*; *Jutfelt et al., 2018*).

In accordance with theoretical expectations, some cold- and warm-adapted marine teleosts can acclimate to minimize deleterious effects of rising ocean temperatures (*Donelson et al., 2011*; *Gienapp et al., 2008*; *Grenchik et al., 2013*; *Norin et al., 2016*; *Somero, 2015*; *Strobel et al., 2012*), maintaining critical processes like growth and swimming performance (*Le Roy et al., 2017*; *Shama and Wegner, 2014*; *Shama et al., 2014*). Other traits, like fecundity, exhibit only limited acclimation capacity in these stenotherms, with concomitant declines in offspring health and survival (*Donelson et al., 2014*; *Donelson et al., 2016*). If critical life-history processes cannot be protected, we are likely to see cascading detrimental impacts on productivity, abundance, and/or whole ecosystem biodiversity across all trophic levels, including ecologically important functional groups and fisheries species (*Brandl et al., 2020*; *Johansen et al., 2015*; *Pearce et al., 2011*; *Rodgers et al., 2018*; *Wernberg et al., 2013*).

Previous seminal work on temperate and freshwater eurythermal species helped clarify which physiological responses are likely to occur during acclimation in stenothermal species (*Angilletta, 2009*; *Barton, 2002*; *Brett, 1971*; *Fry and Hart, 1948*; *Guderley and Blier, 1988*; *Hofmann and Todgham, 2010*; *Madeira et al., 2016*). Elevated temperatures typically increase metabolic processes in all tissues (but see *Brodte et al., 2006*; *Lannig et al., 2005*), subsequently increasing energetic demands (*Fry and Hart, 1948*; *Richter et al., 2010*). In the short-term (i.e. hours, days), processes involved in the oxygen and energy supply may suffer reduced efficiency due to, for example, increased heat-shock protein expression (*Richter et al., 2010*; *Somero, 2012*). In these cases, due to oxygen limitation, anaerobic pathways may be upregulated, typically evidenced by increasing activity of the metabolic enzyme lactate dehydrogenase (LDH; *Jayasundara and Somero, 2013*). Over prolonged time periods (i.e. weeks, months, years), aerobic pathways will be required to sustain performance under elevated energetic demands (*Jayasundara and Somero, 2013*), which can be detected through rising citrate synthase (CS) activity associated with increased mitochondrial density within the tissues (*Fangue et al., 2009*). To increase oxygen delivery to tissues, blood oxygen transport parameters may also be altered, including the number of RBC supplied from cell stores in the spleen (*Lilly et al., 2015*) and increased hemoglobin oxygen binding affinity

(*Jayasundara and Somero, 2013*; *Lilly et al., 2015*). In teleosts, blood oxygen uptake in the gills can be maximized by a reduction in oxygen diffusion distance across the lamellae and/or an increase in total gill surface area available for diffusion (*Bowden et al., 2014*). Finally, the combined adjustments within the tissues, blood, and gills will result in changes at the whole-organism level (*Prosser, 2013*), particularly an increase in resting oxygen uptake rates – a proxy for standard metabolic rate (SMR; *Lefevre, 2016*; *Rao and Bullock, 1954*) and altered maximum oxygen uptake rates during activity, which is a proxy for maximum metabolic rate (MMR; *Pörtner and Farrell, 2008*). The difference between SMR and MMR is aerobic scope (ASc), which indicates the individual's capacity to invest in fitness-enhancing processes beyond basic maintenance (*Eliason et al., 2013*; *Johansen and Jones, 2011*). While many eurythermal species can maintain ASc across a wide temperature span, ASc often declines in stenothermal (i.e. Arctic and tropical) individuals experiencing thermal stress beyond that under which they have evolved (*Drost et al., 2016*; *Franklin et al., 2013*; *Johansen and Jones, 2011*; *Lefevre, 2016*). According to current literature, both the sequence and combination of physiological responses to elevated temperatures remain unclear. Each compensatory mechanism can be energetically costly and/or hinder other fitness-enhancing processes. Individuals must, therefore, balance the tradeoff between the rate of acclimation to thermal stress and the repercussions of each physiological response to overall fitness.

Studies that focus on single physiological responses, single time points, or natural seasonal thermal differences may provide an incomplete snapshot of the capacity for thermal acclimation and the ultimate consequences for fitness and survival under elevated temperatures (*Cossins et al., 1977*; *Madeira et al., 2016*; *Sidell et al., 1973*; *Somero, 2015*). In order to understand the sequence of coordinated physiological responses associated with prolonged thermal stress, an integrative approach across multiple levels of biological organization and time points is required (*Angilletta, 2009*; *Guderley, 1990*; *Johnston and Dunn, 1987*; *Madeira et al., 2016*; *Sidell et al., 1973*; *Somero, 2015*). Currently, we do not have a clear, comprehensive understanding of the sequence or duration of physiological responses – from discrete biochemical mechanisms to whole-animal performance – involved with acclimation to elevated temperatures. These physiological responses will ultimately dictate fitness in a changing environment (*Boyd et al., 2015*; *Horodysky et al., 2015*), either within the lifespan of an individual (e.g. diel variations, seasonal changes, or acute heat-waves) or over generational timespans (e.g. progressive climate change) (*Angilletta, 2009*; *Donelson et al., 2012*; *Schulte et al., 2011*).

This study aimed to identify the key physiological compensatory responses and sequence of responses that occur when warm-adapted stenothermal fishes are exposed to elevated temperatures beyond those for which they have evolved. We focused on two representative stenothermal fishes found in tropical coral reef ecosystems (*Lough, 2012*; *Tewksbury et al., 2008*): the five-lined cardinalfish *Cheilodipterus quinquelineatus* (Apogonidae) and the redbelly yellowtail fusilier *Caesio cuning* (Caesionidae). These species evolved under highly stable thermal regimes, as seasonal water temperatures typically fluctuate less than 4°C across the majority of tropical reefs (*Donner, 2011*; *Lough, 2012*). These species are also separated by more than 100 million years of evolution (*Near et al., 2013*) and differ in most major life-history characteristics, including lifespan (<2 versus >8 years), habitat use (site-attached versus roaming), and mobility (sedentary versus mobile) for *C. quinquelineatus* and *C. cuning*, respectively (*Randall et al., 1997*). We hypothesized that these differences in life history characteristics may alter the physiological responses to unfavorable thermal conditions, with *C. quinquelineatus* resilient to short-term perturbations (due to its sedentary and site-attached life-history) as evidenced by a slower, less substantial acclimation response when compared to *C. cuning* (which could escape unfavorable conditions through its roaming, mobile life history). We compared fish accustomed to current-day maximum temperatures (summer average: 29°C) to fish exposed to conditions typical of acute heating events and in line with projected future climate change conditions (+3.0°C) (*Garrabou et al., 2009*; *Hughes et al., 2017*; *IPCC, 2013*; *Pörtner et al., 2019*). Thermal acclimation was assessed weekly over a 5-week period using a comprehensive set of 18 hematological and cardiorespiratory parameters. These parameters included biochemical pathways within tissues and blood, blood oxygen transport and gill morphology, as well as whole animal metabolism and body condition (see *Table 1* for full list). Based on previous reports of the timing and duration of physiological responses of stenothermal fishes to elevated temperatures (e.g. *Madeira et al., 2016*; *Sidell et al., 1973*; *Somero, 2015*), we hypothesized that it would

**Table 1.** Acclimation responses of 18 hematological and cardiorespiratory parameters.

| Type | Parameter | Definition |
|---|---|---|
| Muscle + Gill | Citrate synthase activity (CS) | An exclusive marker of the mitochondrial matrix and a marker of mitochondrial density in tissues |
| | Lactate dehydrogenase (LDH) | An enzyme involved in anaerobic energy production |
| Spleen | Spleen somatic index (SSI) | The relative spleen to body mass, used to assess release of red blood cell stores into the blood stream |
| | Spleen [Hb] | An indicator of red blood cell production within the spleen |
| Blood | Mean corpuscular hemoglobin content (MCHC) | Hemoglobin concentration in red blood cells, indicative of blood oxygen carrying capacity |
| | Hematocrit (Hct) | The ratio of red blood cells to the total volume of blood |
| | Hemoglobin (Hb) | The protein responsible for transporting oxygen in the blood |
| | Whole blood lactate | Lactic acid appears in the blood as a result of anaerobic metabolism |
| | Whole blood glucose | Used to support the metabolic pathways in the mitochondria and cytoplasm |
| Gill | Lamellar perimeter | The perimeter of a cross-section of the lamellae measured histologically as a proxy for total diffusible surface area for $O_2$ transport over the lamellae. |
| | Lamellar width | The histological total diameter of the lamellae epithelium and capillary. Lamellar width is here used to indicate the epithelial thickness of the lamellae (i.e. diffusion distance for $O_2$). |
| | Epithelial thickness | A measure of the diameter of the epithelia on the lamellae |
| Whole body metabolism | Standard metabolic rate (SMR) | Baseline oxygen consumption required to maintain bodily functions. |
| | Maximum metabolic rate (MMR) | Maximal oxygen consumption |
| | Aerobic scope (ASc) | The difference between MMR and SMR, indicating the maximal $O_2$ available for activity. |
| Whole body condition | Fulton's K condition factor | Length-mass relationship used to estimate health of an individual |

take a minimum of 3 weeks for the putative 'slower acclimating species' (*C. quinquelineatus*) to stabilize all hematological and cardiorespiratory parameters following elevated temperature exposure.

## Results

### Weeks 0–1: initial thermal stress responses

Relative to controls, *C. cuning* showed changes in blood and muscle parameters within the first week of exposure to elevated temperatures. Blood glucose levels in *C. cuning* more than doubled from weeks 0 to 1, from 1.93 to 4.65 mM ($p_{p.c.week1}$ = 0.003; *Figure 1*) (p.c.week1 denotes a planned comparison [p.] between control [c.] and week one individuals. See Materials and methods for details). In the pectoral muscle of *C. cuning*, the anaerobic metabolic enzyme, LDH, declined precipitously in week 0 compared to controls, from 1.36 to 0.61 mM ($p_{p.c.week0}$ = 0.003; *Figure 2*). Conversely, muscle LDH in *C. quinquelineatus* was consistent with control levels from weeks 0 to 1 ($p_{p.c.week0}$ > $P_{cutoff}$, $p_{p.c.week1}$ > $P_{cutoff}$; *Figure 2*).

Both *C. cuning* and *C. quinquelineatus* exhibited changes in metabolic oxygen transport characteristics, demand, and/or aerobic capacity immediately following exposure to elevated temperature. Relative to controls, *C. cuning* exhibited greater lamellar width ($p_{p.c.week1}$ = 0.002; *Figures 3* and *4*) and higher SMR ($p_{p.c.week0}$ = 0.021; *Figure 5*) within 1 week of exposure to the elevated temperature treatment. However, only moderate, non-significant increases in MMR ($p_{p.c.week0}$ > $P_{cutoff}$) and ASc ($p_{p.c.week0}$ > $P_{cutoff}$; *Figure 5*) were found. Conversely, *C. quinquelineatus* showed no acute changes to gill structure (*Figures 3* and *4*) and only a moderate, non-significant increase in SMR ($p_{p.c.week0}$ > $P_{cutoff}$; *Figure 5*). However, both MMR and ASc were significantly higher for *C. quinquelineatus* in the first week of temperature treatment, relative to controls (MMS: $p_{p.c.week0}$ <0.001, $p_{p.c.week1}$ <0.001; ASc: $p_{p.c.week0}$ <0.001, $p_{p.c.week1}$ <0.001; *Figure 5*). After 1 week of acclimation, SMR showed no significant difference from control levels in either species (*Figure 5*). Importantly, while no mortality was recorded for *C. cuning* in weeks 0 and 1, *C. quinquelineatus* showed a 79% and 20% mortality after exhaustive exercise in weeks 0 and 1, respectively (*Appendix 1—figure 1*).

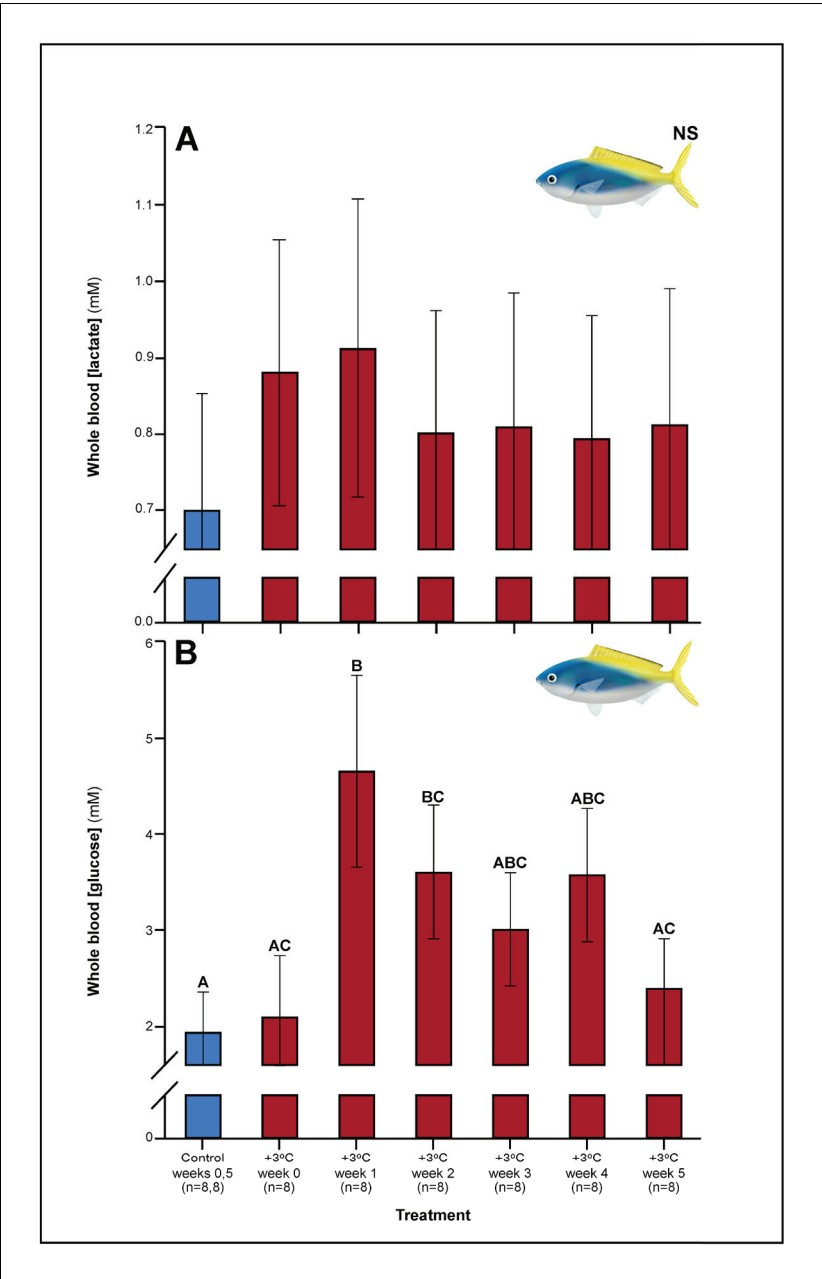

**Figure 1.** Effect of 5 weeks' exposure to elevated temperatures (+3.0°C) on whole blood. (A) Lactate and (B) glucose concentrations in *Caesio cuning*. The first blue column in each figure illustrates the control (29.0°C). Letters above columns indicate significant differences between treatment groups, determined through multiple comparisons *post-hoc* testing (based on linear mixed-effects model analysis). 'NS' denotes that the model did not indicate any significant effects of temperature. Error bars are s.e.m., and numbers in parentheses below each category on the x-axis denote sample sizes for each group.

## Weeks 2 –3: delayed acclimation responses

A number of traits exhibited delayed acclimation responses 2–3 weeks after exposure to elevated temperature. In *C. cuning*, blood glucose levels remained elevated until week 2 following high-temperature treatment ($p_{p.c.week2}$ = 0.029); however, levels did not vary significantly different from controls from week 3 onwards (*Figure 1*). In week 3, *C. quinquelineatus* showed a peak in LDH activity in the pectoral muscle ($p_{p.c.week3}$ = 0.021; *Figure 2*), and spleens contracted by 56.2% (as measured through spleen somatic index, SSI; $p_{p.c.week3}$ <0.001; *Figure 6*), when compared to controls.

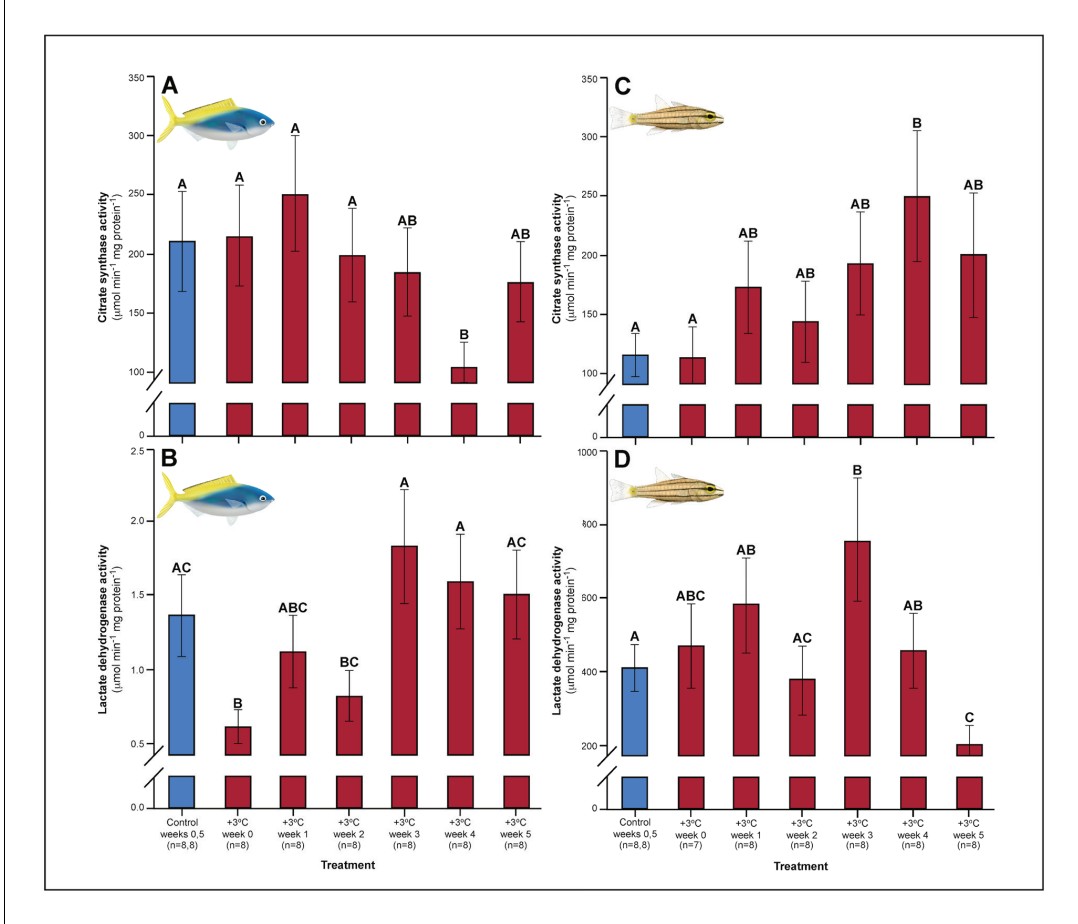

**Figure 2.** Effect of 5 weeks' exposure to elevated temperature (+3.0°C) on citrate synthase (CS) and lactate dehydrogenase (LDH) enzyme activity in the pectoral muscle of *Caesio cuning* (A and B) and *Cheilopterus quinquelineatus* (C and D). The first blue column in each figure illustrates the control (29.0°C). Letters above columns indicate significant differences between treatment groups, determined through multiple comparisons *post-hoc* testing (based on linear mixed-effects model analysis). Error bars are s.e.m., and numbers in parentheses below each category on the x-axis denote sample size.

Gill morphology in both species also shifted in weeks 2 and 3. *Caesio cuning* continued to show increased lamellar width through weeks 2 and 3 ($p_{p.c.week2}$ <0.001; $p_{p.c.week3}$ <0.001; *Figure 4*). This was augmented in week 3 by a significant increase in total diffusible lamellar perimeter ($p_{p.c.week3}$ <0.001; *Figure 3*). For *C. quinquelineatus*, initial epithelial thickness and lamellar width were maintained (*Figure 4*), but showed a significant decrease in total lamellar perimeter in week 2 ($p_{p.c.week2}$ <0.001; *Figure 3*).

Metabolic rate also continued to change during this time period in both species. The acclimation responses initiated by *C. cuning* culminated in a peak in whole animal MMR and ASc after 3 weeks' exposure to elevated temperatures when compared to controls (MMR, $p_{p.c.week3}$ = 0.013; ASc, $p_{p.c.week3}$ = 0.017; *Figure 5*). Conversely, MMR remained elevated through week 2 for *C. quinquelineatus* ($p_{p.c.week2}$ = 0.002; *Figure 5*). This species' ongoing morphological and biochemical adjustments came with a brief reduction in both SMR and MMR below control values in week 3 (SMR: $p_{p.c.week3}$ = 0.002; MMR: $p_{p.c.week3}$ = 0.015), leading ASc to return to control levels in this species (*Figure 5*). Again, *C. quinquelineatus* exhibited 56% and 38% mortality after exhaustive exercise during weeks 2 and 3, respectively (*Appendix 1—figure 1*), while no mortality was recorded for *C. cuning* throughout the study.

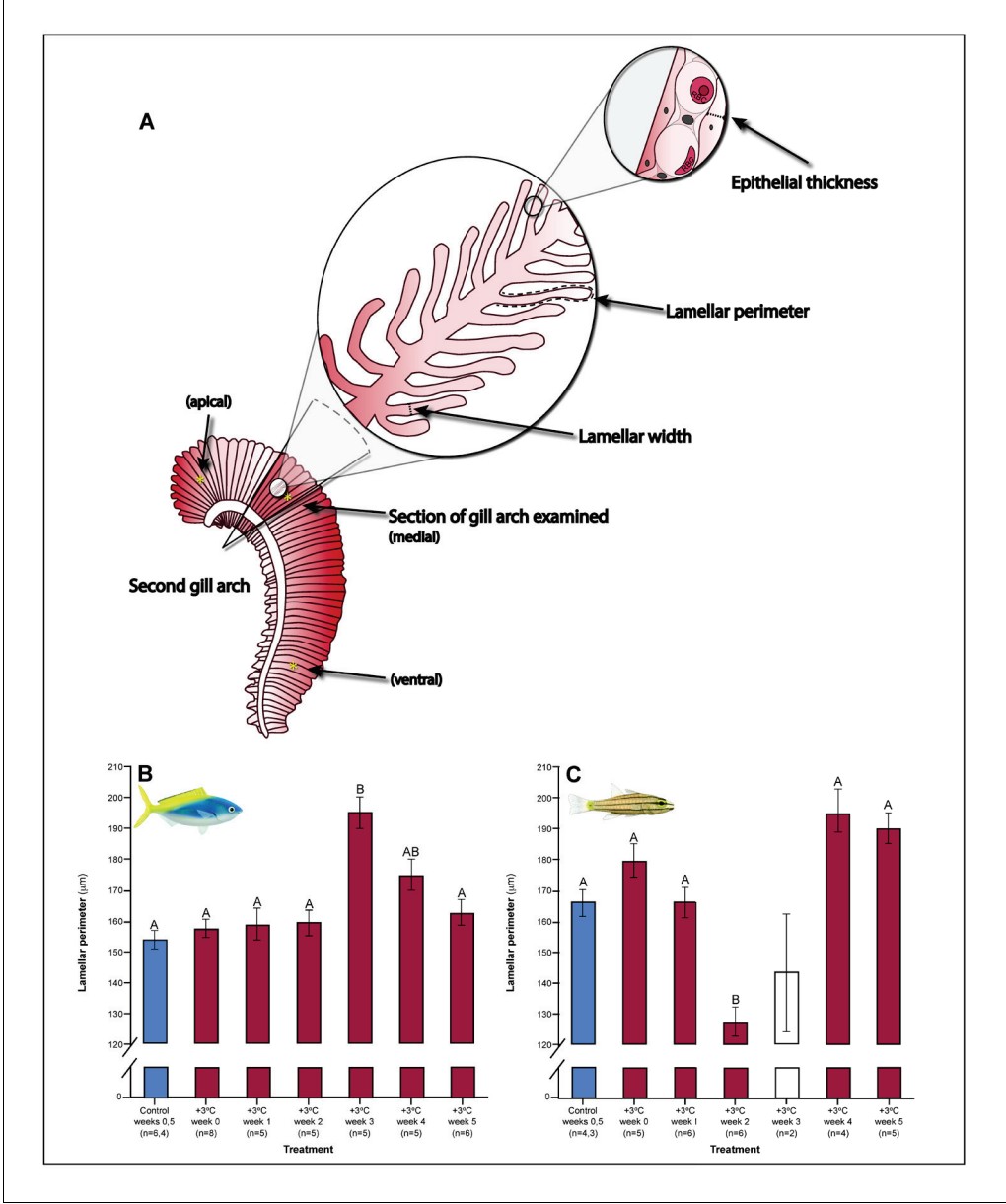

**Figure 3.** Gill parameters examined throughout the five weeks' exposure to elevated temperature (+3.0 ℃). (**A**) Illustration of gill parameters examined. The response of lamellar perimeter to elevated changed through time (in mm) in *Caesio cunning* (**B**) and *Cheilopterus quinquelineatus* (**C**). The first blue column in each figure illustrates the control (29.0 ℃). Letters above columns indicate significant differences between treatment groups, determined through multiple comparisons post-hoc testing (based on linear mixed-effects model analysis). The white bar denotes data excluded from analyses due to low n, but shown for clarity. Error bars are s.e.m., and numbers in parentheses below each category on the x-axis denote sample size.

## Weeks 3–5: stabilization of acclimation responses

The succession of physiological parameters began plateauing after 3–5 weeks of continuous exposure to elevated temperatures. In *C. cuning*, CS activity in the gill and pectoral muscle was one of the last parameters to respond to elevated temperatures, as both traits declined to a significant minimum in week 4 (Muscle: $p_{p.c.week4}$ = 0.011; Gill: $p_{p.c.week4}$ <0.001) before returning to control levels in week 5 ($p_{p.c.week5}$ >0.5 for both; *Figure 2*, *Appendix 1—figure 2*). *Caesio cuning* also required 4 weeks for all morphological gill parameters to return to control values ($p_{p.c.week4}$ > $P_{cutoff}$), following

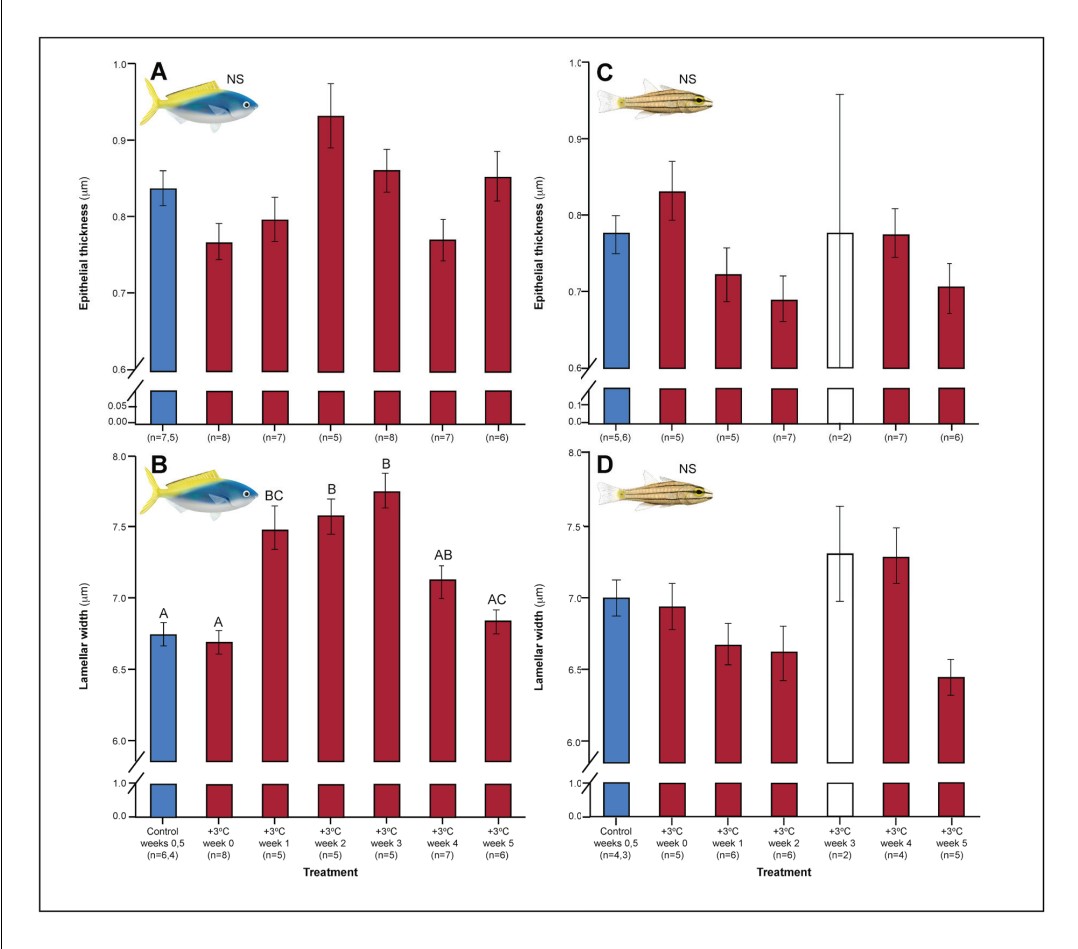

**Figure 4.** Effect of 5 weeks' exposure to elevated temperatures (+3.0°C) on epithelial thickness and lamellar width (both in µm) in the gills of *Caesio cuning* (A and B) and *Cheilopterus quinquelineatus* (C and D). The first blue column in each figure illustrates the control (29.0°C). Letters above columns indicate significant differences between treatment groups, determined through multiple comparisons *post-hoc* testing (based on linear mixed-effects model analysis). The white bar denotes data excluded from analyses due to low n, but shown for clarity. 'NS' denotes that the model did not indicate any significant effects of temperature. Error bars are s.e.m., and numbers in parentheses below each category on the x-axis denote sample size.

increased lamellar width in weeks 1–3 and increased lamellar perimeter in week 3 (*Figures 3* and *4*). Conversely, *C. quinquelineatus* showed no change in gill CS activity but exhibited a peak in CS activity in the pectoral muscle in week 4 ($p_{p.c.week4}$ = 0.004; *Figure 2*) and declining muscle LDH activity from weeks 3 to 5 ($p_{p.c.week5}$ = 0.016; *Figure 2*). In both species, MMR and ASc both returned to control values in week 4 (*Figure 5*). However, in *C. quinquelineatus*, this was accompanied by a 38.5% mortality rate after exhaustive exercise in weeks 3 and 4 (*Appendix 1—figure 1*).

## Past week 5: ongoing acclimation

Several parameters had yet to stabilize in either species after 5 weeks' exposure to elevated temperatures. For *C. cuning*, spleen [Hb] exhibited a non-significant trend ($p_{p.c.week5} > P_{cutoff}$) to increase over the entire 5-week period culminating in a significant peak in week 5 relative to week 0 ($p_{p.week0.week5}$=0.007), indicative of ongoing compensatory mechanisms for blood oxygen transport (*Figure 6*). While *C. quinquelineatus* showed no significant change in spleen [Hb] over the 5-week exposure period, this species did exhibit a significant reduction in spleen somatic index ($p_{p.c.week5}$ <0.001, *Figure 6*) from week 3 onwards, through the end of the 5-week exposure period. This sequence of physiological responses over 5 weeks (and ongoing acclimation) is summarized in *Figure 7*.

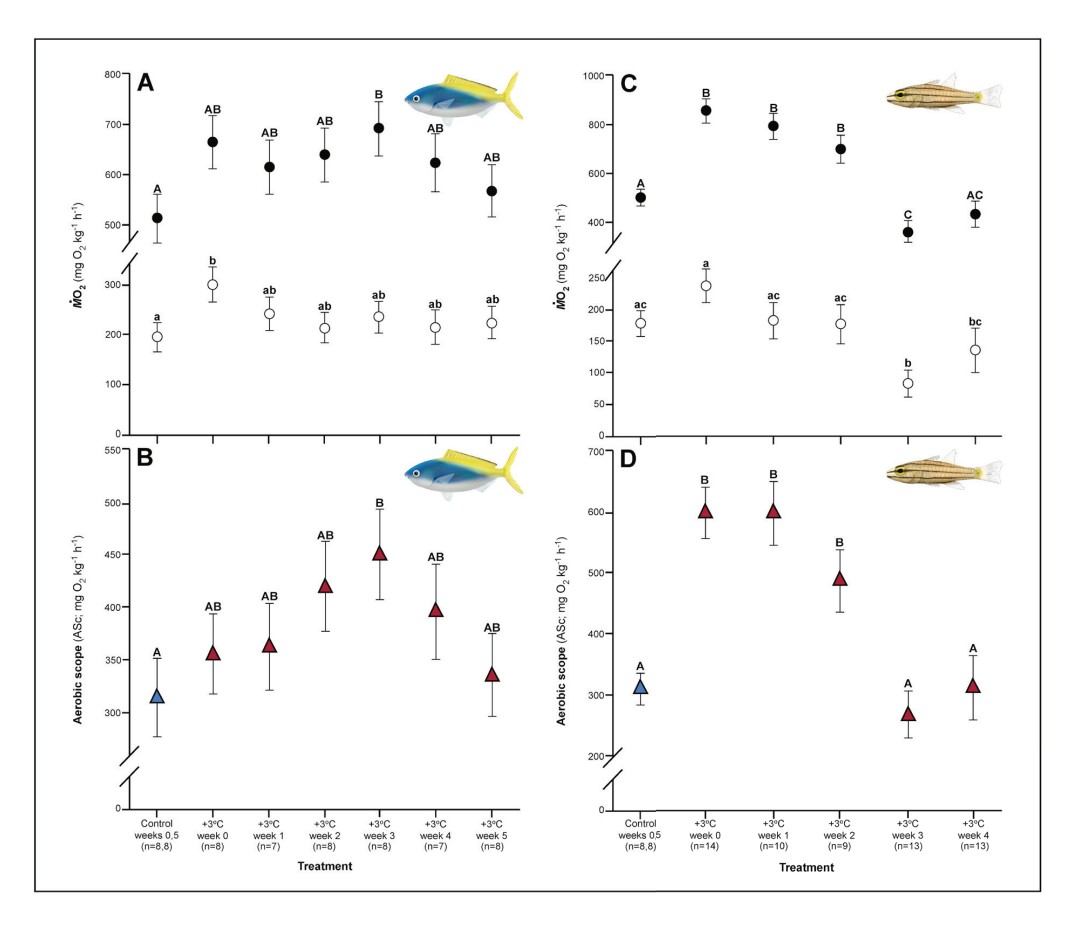

**Figure 5.** Effect of 5 weeks' exposure to elevated temperatures (+3.0°C) on metabolic performance of *Caesio cuning* (A and B) and *Cheilopterus quinquelineatus* (C and D). Estimates of standard (white circles) and maximum (black circles) metabolic rates (A and C) and aerobic scope (ASc = triangles, B and D) are illustrated. The first blue data point in panels B and D represent the control (29.0°C). Letters above data points indicate significant differences between treatment groups, determined through multiple comparisons *post-hoc* testing (based on linear mixed-effects model analysis). Error bars are s.e.m., and numbers in parentheses below each category on the x-axis denote sample size.

## Non-responsive parameters

Several parameters were unaltered by elevated temperature through time. Neither species exhibited significant plasticity in gill LDH activity over the 5-week exposure period (*Appendix 1—figure 2*). Additionally, although spleen [Hb] increased over the 5-week exposure to elevated temperatures in *C. cuning* (*Figure 6*), we found no indication of new RBCs being released into circulation over the first 5 weeks' exposure, as there were no significant differences in hematocrit, whole blood [Hb], MCHC, or spleen somatic index between control and +3.0°C-exposed individuals ($p_{p.c.week5} > p_{cutoff}$; *Figure 6*, *Appendix 1—figure 2*). *Caesio cuning* also showed no change in blood [lactate] from control levels at any time point (*Figure 1*), suggesting that this species was not relying heavily on anaerobic energy production to the point that lactate was detectable in the plasma. Individuals from both species and all treatments also maintained Fulton's K condition factor (i.e. body condition, *Appendix 1—figure 3*), suggesting that the physiological changes detected in +3.0°C-exposed individuals were due to changing temperature conditions rather than waning health under laboratory confinement.

## Discussion

Understanding the impacts of global change on species fitness and resilience requires a comprehensive, integrative approach across multiple levels of biological organization and time points

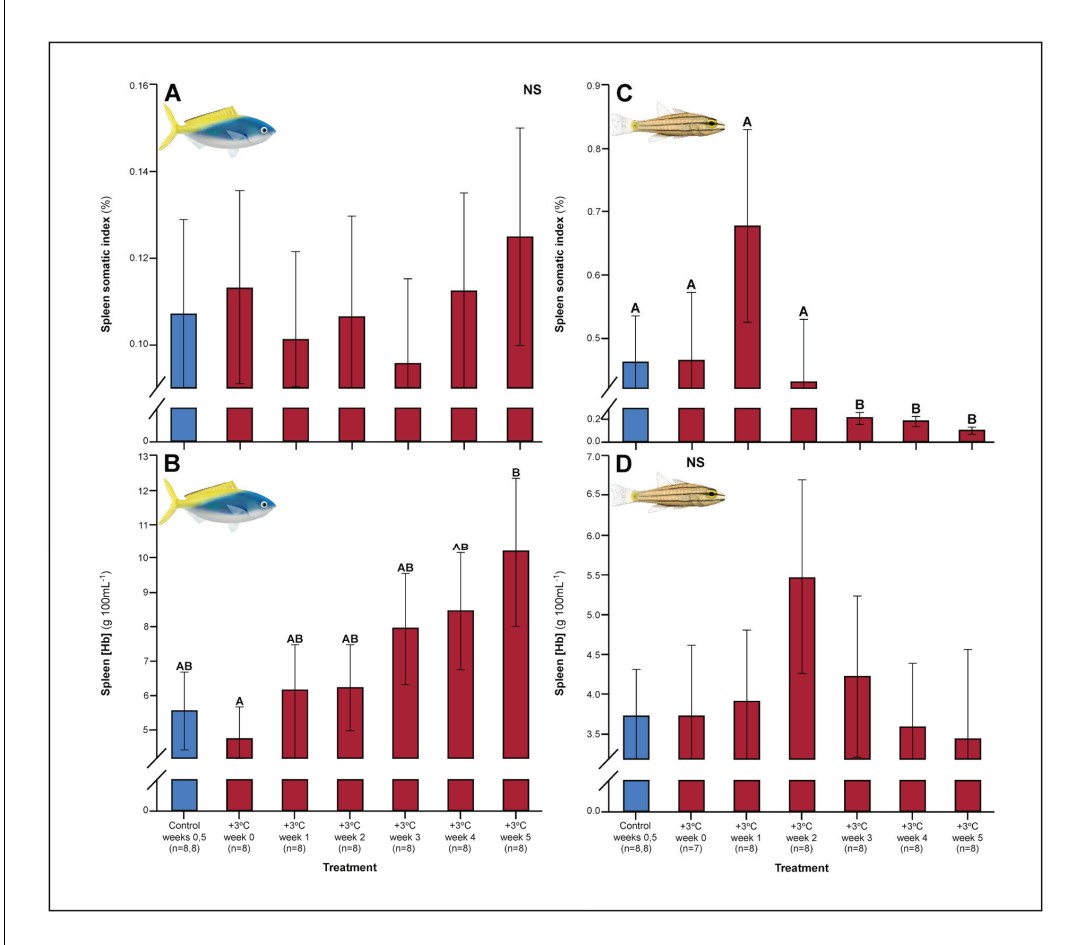

**Figure 6.** Effect of 5 weeks' exposure to elevated temperatures (+3.0°C) on spleen somatic index and spleen hemoglobin concentration of *Caesio cuning* (A and B) and *Cheilopterus quinquelineatus* (C and D). The first blue column in each figure illustrates the control (29.0°C). Letters above columns indicate significant differences between treatment groups, determined through multiple comparisons *post-hoc* testing (based on linear mixed-effects model analysis). 'NS' denotes that the model did not indicate any significant effects of temperature. Error bars are s.e.m., and numbers in parentheses below each category on the x-axis denote sample size.

(*Cossins et al., 1977*; *Sidell et al., 1973*; *Somero, 2015*). Here, we provide the first comprehensive assessment of the key compensatory mechanistic responses and sequence of events that occur when warm-adapted stenothermal fishes are exposed to elevated temperatures above those for which they have evolved. Elevated temperature resulted in discrete responses in the gills, blood, spleen, and muscle across 13 separate hematological and cardiorespiratory parameters (see *Table 1* for details). As hypothesized, the specific acclimation responses and duration of events differed among the study species, potentially due to divergent evolution and ecology. *Caesio cuning* exhibited rapid responses to thermal stress, with nearly immediate changes detected in gill morphology and blood parameters. *Cheilodipterus quinquelineatus*, conversely, largely exhibited a delayed response in all parameters measured, potentially contributing to the much higher (>50%) mortality seen across the 5-week heating event. Importantly, we identified seven conserved physiological parameters across both species that may be useful as biomarkers for evaluating the progression of acclimation in thermally sensitive teleosts, and which may greatly improve our understanding and projections of ongoing climate change threats and disturbances.

## Initial stress responses

For ectotherms, rising temperature causes an exponential increase in the energetic costs associated with maintaining bodily functions (i.e. Q10, *Rao and Bullock, 1954*). Accordingly, short-term

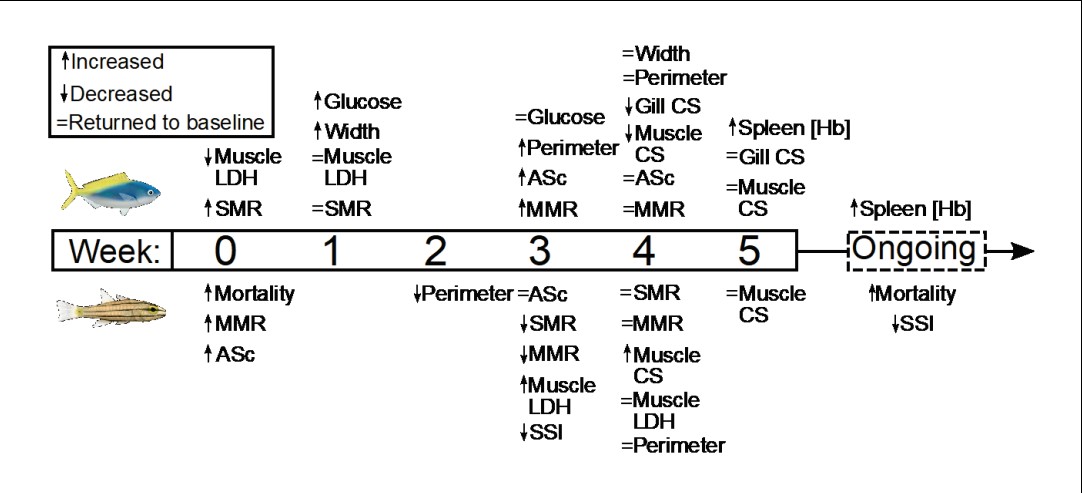

**Figure 7.** Summary of the timing and duration of the physiological responses of two stenothermal fishes (Caesio cuning, top, and Cheilodipterus quinquelineatus, bottom) to a five-week acute heating event. Summary of the timing and duration of the physiological responses of two stenothermal fishes (*Caesio cuning*, top, and *Cheilodipterus quinquelineatus*, bottom) to a 5-week acute heating event, including responses in mortality, whole organism demand (standard metabolic rate, SMR), whole organism supply (maximum metabolic rate, MMR, and aerobic scope, ASc), tissue demand (citrate synthase, CS, and lactate dehydrogenase, LDH, enzyme activity in the gill and muscle tissue), and tissue supply (lamellar width and perimeter, blood glucose, spleen [Hb], and spleen somatic index, SSI).

exposure (≤1 week) to elevated temperatures elicited primary stress responses in the blood, gills, and whole animal metabolism, which is consistent with a rapid increase in total energy demand (e.g. *Angilletta, 2009*; *Brett, 1971*; *Fry and Hart, 1948*; *Guderley and Blier, 1988*). *Caesio cuning* appeared to increase energy delivery by increasing blood glucose levels, which allows enhanced energy production via aerobic oxidative phosphorylation and glycogenolysis in the liver (*Jayasundara et al., 2013*). In order to increase aerobic energy production, teleosts typically also require increased oxygen transport at the gills for delivery to tissues and organs (e.g. *Jensen et al., 1993*). Elevated temperatures will increase rates of diffusion across the gill epithelium, and our results suggest that *C. cuning* increased blood flow through the gills for increased oxygen transport (denoted by increased lamellar width). Accordingly, SMR rose immediately by 53.5% in *C. cuning* following exposure to elevated temperatures. Interestingly, *C. quinquelineatus* did not appear to similarly increase energy supply to the organs, as there was no immediate increase in lamellar width. This species also showed a staggering 79% mortality following exercise within the first week, suggesting that the limited acclimation response may have been associated with systemic organ failure in a high proportion of individuals.

## Prolonged acclimation responses

Secondary acclimation processes were initiated after 2–3 weeks of continuous exposure to elevated temperatures. The sequential responses in tissue enzyme activity, SSI, and gill morphology can be linked to increased RBC concentration, enhanced oxygen transport, and improved gill oxygen uptake (*Jayasundara et al., 2013*; *Windisch et al., 2011*). These concurrent adjustments likely contributed to the measured increase in MMR and ASc through at least the second week and thereby reduced trade-offs between energy production and demand within the tissues.

Although the observed adaptive remodeling in gill structure would allow for enhanced oxygen acquisition (*Sollid and Nilsson, 2006*), changes in gill morphology can lead to trade-offs with other physiological processes. Gill structure, in terms of diffusion distance (i.e. epithelial thickness), total surface area (i.e. lamellar perimeter), and permeability (i.e. lamellar width), dictates both respiration (i.e. in terms of the diffusion rate of carbon dioxide and oxygen between the gills and surrounding water) and ion exchange for osmoregulation (*Sardella and Brauner, 2007*). However, the osmorespiratory compromise in fish posits that gill structural characteristics that promote effective respiration are contrary to those that support efficient osmoregulation (*Nilsson, 1986*). *Caesio cuning*, in particular, showed increased surface area (as evidenced by the significantly higher lamellar perimeter

and width) 2–3 weeks after the initial spike in temperature, which would increase oxygen uptake capacity at the expense of reduced control of ion exchange in the gills. Following challenges (e.g. chasing by a predator, environmental factors), these gill changes are likely to impact osmotic homeostasis (*Sardella and Brauner, 2007*), particularly on shallow, inshore habitats where salinity can vary diurnally with the tide, storms, and runoff (*Lough et al., 2015*).

After 3–5 weeks of continuous exposure to elevated temperatures, multiple systemic morphological and biochemical parameters started stabilizing at levels that appeared to reduce trade-offs between competing biological functions. Indicators of anaerobic and aerobic energy metabolism within the blood and tissues peaked by week 4 (specifically LDH, MMR, ASc, lamellar perimeter and width, and muscle and gill CS). Most parameters subsequently returned to control values in weeks 4–5, consistent with a steady decreasing need for anaerobic metabolism and increasing aerobic energy production. These trends suggest that adjustments to sustain increased energy production required at least 4–5 weeks of continuous exposure to elevated temperatures to plateau (see *Table 1* for duration of individual parameters). However, in both species, pathways relevant for RBC production and storage exhibited signs of on-going recovery beyond week 5, with SSI still depleted in *C. quinquelineatus* and rising spleen [Hb] in *C. cuning*.

## Divergent acclimation pathways with same endpoints relates to species evolution and life history

We found two divergent strategies for maintaining oxygen transport and energy production during prolonged exposure to elevated temperatures. *Caesio cuning* exhibited a more immediate and comprehensive series of responses to thermal stress, including increasing splenic hemoglobin concentrations (i.e. RBC production), progressive changes in gill structure, altered aerobic and anaerobic enzymatic activity, and elevated blood glucose. These physiological responses can be associated with increasing oxygen transport in response to rising metabolic maintenance costs at high temperature (*Seebacher et al., 2015*; *Sollid and Nilsson, 2006*). Importantly, the combined physiological adjustments within *C. cuning* appeared to allow this species to return maximal oxygen transport to control levels within 5 weeks, and allowed them to maintain a high survival rate despite thermal stress. In contrast, *C. quinquelineatus* appeared to have a very limited capacity for rapid acclimation, as thermal stress severely compromised survival. As hypothesized for this species, plasticity in physiological and morphological traits appeared to be delayed, with few changes observed until 2–3 weeks following exposure to elevated temperatures, at which point plasticity in enzyme activity, gill structure, and SSI were finally observed. These delays likely contributed to the notably high mortality rate observed in the first 2 weeks in the high-temperature treatment group for this species (*Li et al., 2015*), with 20–79% mortality rates following exercise during this time period.

These two contrasting acclimation responses (immediate vs. delayed) may be related to the life history strategies of each species. Notably, *C. cuning* is a roaming species that utilizes a high proportion of aerobic red musculature and exhibits greater aerobic metabolic performance to move rapidly among habitats seeking foraging opportunities (*Randall et al., 1997*). As a result, this species is equipped to leave areas of unfavorable localized biophysical conditions, for instance, by swimming deeper when temperatures in the shallow waters become too warm (*Randall et al., 1997*). By comparison, *C. quinquelineatus* is a sedentary, highly site-attached and territorial species, remaining between coral branches during daytime and foraging in the water column close to shelter at night (*Randall et al., 1997*). For swimming, *C. quinquelineatus* relies almost exclusively on anaerobic white glycolytic muscle (*Stickland, 1975*) and may regularly be exposed to short fluctuations of temperatures in its sheltered, shallow water habitats (e.g. during low tide) (*Biro et al., 2010*). Consequently, in light of these life history strategies, it is logical for *C. cuning* to immediately initiate metabolically costly compensatory mechanisms in situations where a stressor cannot be avoided, while *C. quinquelineatus* would benefit from delaying similar processes given the frequency with which they are exposed to short-term perturbations. Unfortunately, while delayed responses may be beneficial for short-term and relatively benign fluctuations in temperature, this strategy is unlikely to be advantageous under prolonged thermal stress and may come at the expense of fitness-enhancing processes. Indeed, SMR seemed to temporarily reduce below control levels in *C. quinquelineatus* in week 3, potentially indicating incipient organ failure or a shutdown of non-critical functions, such as reproductive organs. More than 50% of all *C. quinquelineatus* died over the course of the 5-week acclimation period following exercise; whereas, no mortality was observed in *C. cuning*, potentially

highlighting that we only saw acclimation responses in the most thermally resilient subset of the *C. quinquelineatus* population.

The devastating impacts of climate change on marine resources reinforces the urgent need for cross-disciplinary studies addressing the timescales, mechanisms, and limitations of thermal responses in ectothermal marine species (*Audzijonyte et al., 2019*; *Jutfelt et al., 2018*). The GOLT hypothesis asserts that fishes are unable to make necessary morphological adjustments to gill tissues to meet increased oxygen demands under elevated temperatures (*Pauly, 2019*) and has been proposed as a unifying explanation for an observed reduction in size and mass of fishes under climate change (*Pauly and Cheung, 2018*). Here, we found clear signs of adaptive remodeling of the gill perimeter and width in both of the species tested, resulting in increased lamellar surface area after 1–3 weeks of continuous exposure to elevated temperatures. As such, the responses reported here do not conform to the GOLT. Instead, these data provide further support to the counter argument in *Lefevre et al., 2017*, *Lefevre et al., 2018*, which states that gill surface area can increase to support metabolic needs and that temperature-driven changes in size and mass must be explained by other mechanisms (*Lefevre et al., 2018*). Our results do, however, support the idea that cardio-respiratory transport and tissue demand are primary determinants of an organism's performance under ocean warming (see extended discussions in *Ern, 2019*; *Portner, 2014*; *Sandblom et al., 2016*), as an increase in total energy supply appeared to support *C. cuning* through the 5-week acclimation period. whereas a lack of aerobic energy availability was associated with mass mortality in *C. quinquelineatus.*

## Useful biomarkers of thermal stress and ongoing acclimation

Seven biomarkers stood out as useful, conserved indicators to assess the degree and progress of thermal acclimation. These include muscle CS activity, as an indication of changes in aerobic metabolism and mitochondrial density (*Jayasundara et al., 2013*; *Windisch et al., 2011*); blood glucose and muscle LDH activity, as indicators of anaerobic energy production (*Jayasundara et al., 2013*; *Windisch et al., 2011*); splenic RBC stores (SSI and Spleen [Hb]), denoting altered oxygen transport capacity (*Ken-Ichi, 1988*), and lamellar perimeter and width, illustrating changes to oxygen uptake capacity over the gills (*Sollid and Nilsson, 2006*). These seven parameters responded to short-term and prolonged exposure to elevated temperatures in one or both of our study species, and all subsequently returned to control values, except for splenic RBC stores. Although never previously examined in such detail, similar responses have been suggested in other non-warm adapted, thermally-sensitive species during prolonged exposure to elevated temperatures (*Jayasundara et al., 2013*; *Windisch et al., 2011*). The Antarctic notothenioid (*Trematomus bernacchii*) and eelpout (*Pachycara brachycephalum*), for example, both appear to increase CS and LDH activities in response to rising energetic needs (*Jayasundara et al., 2013*; *Windisch et al., 2011*). These responses were attributed to both a shift in the primary metabolic fuel from lipids to carbohydrates, as carbohydrates can sustain anaerobic metabolism (*Jayasundara et al., 2013*), and a decrease in ATP-generating capacity, either due to limitations in oxygen delivery or impaired oxidative phosphorylation due to mitochondrial failure (*Jayasundara et al., 2013*; *Windisch et al., 2011*). Our data provide new insight to support these findings for warm- and cold-adapted stenotherms alike. Similar responses in oxygen transport capacity and gill structure have also previously been recorded in some temperate marine and tropical freshwater fishes (*Chapman et al., 2000*; *Sollid and Nilsson, 2006*). Populations of African cichlids can increase lamellar surface area to facilitate greater oxygen transport (*Chapman et al., 2000*; *Schaack and Chapman, 2003*), highlighting that the acclimation responses identified here may, in fact, be useful biomarkers in a range of tropical species. Due to differences among species, however, our data highlight that it is critical to examine several biomarkers together to more accurately ascertain whether acclimation is still ongoing at a given time point.

## Conclusions and recommendations

A mechanistic understanding of the rates, processes, and limitations of acclimation to global warming is critically important to understand species resilience into the future and to improve the accuracy of fisheries and ecosystem projections. Our study provides the first comprehensive, mechanistic investigation of the thermal acclimation processes of tropical stenothermal fishes and the rates at which they occur. Some parameters, such as SMR, may adjust within as little as 2 weeks. However,

many other physiological parameters, including ASc and splenic RBC stores, appear to require significantly longer exposure periods to adjust. These results suggest that studies relying on shorter acclimation periods of less than 5 weeks may not capture the complete acclimation or performance capacity of a species following an acute heating event.

## Materials and methods

### Study species and maintenance

Two species of tropical coral reef fishes were collected from locations in the northern Great Barrier Reef using barrier nets, hand nets, and a mild clove oil anaesthetic solution under Marine Parks Permit #G10/33239.1. The five-lined cardinalfish, *C. quinquelineatus* (n = 143 fish; 1.48 ± 0.08 g, body mass ± SE), was collected from Sudbury Reef (17.00°S, 146.08°E) in the Cairns region in March-April 2014. The redbelly yellowtail fusilier *C. cuning*, (n = 110 fish; 22.20 ± 1.03 g, body mass ± SE), was collected from the reefs surrounding Lizard Island (14.67°S, 145.46°E) in December 2013. In both instances, water temperature was within 1°C of control. Fishes were transported to the Marine and Aquaculture Research Facilities Unit (MARFU) at James Cook University in Townsville, Queensland, Australia.

All fishes were maintained in aquaria for a minimum of 14 days before acclimation trials began. Fishes were maintained with a continuous supply of recirculated, filtered, aerated, and UV-sterilized sea water and fed to satiation daily with aquaculture pellets (NRD pellets, INVE Aquaculture, Salt Lake City, USA) and hatched *Artemia* spp. under a 12 hr:12 hr light:dark photoperiod. Individuals of each species were randomly assigned to the control or high-temperature (+3.0°C) treatment and either assessment of metabolic performance (using respirometry protocols) or tissue analyses (see *Table 2* for sample sizes by treatment). To account for size differences between the two study-species, all experimental setups and holding tanks were optimized for body size (body mass, standard length, fish density, and total biomass). All *C. cuning* individuals (i.e. both those assigned to respirometry protocols and tissue analyses) were housed in 400 L round aquaria (diameter: 1.2 m, depth: 50 cm), at a density of approximately 1.4 g L$^{-1}$ seawater. One week prior to experimentation, individual *C. cuning* that were assigned to respirometry trials were implanted with a unique visible implant elastomer (VIE) tag (Northwest Marine Technology, Inc, Shaw Island, USA) subcutaneously below the dorsal fin, which allowed for individual recognition through time with little to no deleterious impacts to the animal (*Hoey and McCormick, 2006*). For *C. quinquelineatus*, given cardinalfishes sensitivity to handling stress (*Nilsson et al., 2010*; *Rummer et al., 2014*), individuals assigned to respirometry trials were each placed in their own individual transparent 3 L aquaria adjacent to conspecifics, in order to allow for individual identification through time while minimizing risk of stress-induced mortality. Fish used for tissue analyses were housed in 9L aquaria (W:22.0 cm x H:22.5 cm x L:22.0 cm) at a density of approximately 1 g L$^{-1}$ seawater. All animal care and experimental procedures complied with those framed by the James Cook University Animal Ethics Committee (Permit A2089, approved for this study).

**Table 2.** Sample size of each parameter across species and exposure week.
For histological samples, each Individual was used for both tissue and gill analyses. For respirometry (due to ethical requirements to minimize sample numbers), we used a mixed-staggered repeated measures design in which each individual was tested twice where possible, but with a 3-week separation between each trial (i.e. Week 0 and Week 3, Week 1 and Week 4, and Week 2 and Week 5).

| Species | Parameter | Ctrl Week 0 | Crtl Week 5 | Week 0 | Week 1 | Week 2 | Week 3 | Week 4 | Week 5 |
|---|---|---|---|---|---|---|---|---|---|
| *C. quinquelineatus* | Tissue | 8 | 8 | 7 | 8 | 8 | 8 | 8 | 6 |
| *C. quinquelineatus* | Gill (thickness, perimeter, width) | 5, 4, 4 | 6, 3, 3 | 5, 5, 5 | 5, 6, 6 | 7, 6, 6 | 2, 2, 2 | 7, 4, 4 | 6, 5, 5 |
| *C. quinquelineatus* | Respirometry / mortality | 8 | 8 | 14 | 10 | 9 | 13 | 13 | 0 |
| *C. cuning* | Tissue / blood | 8 | 8 | 8 | 8 | 8 | 8 | 8 | 8 |
| *C. cuning* | Gill (thickness, perimeter, width) | 7, 6, 6 | 5, 4, 4 | 8, 8, 8 | 7, 5, 5 | 5, 5, 5 | 8, 5, 5 | 7, 5, 7 | 6, 6, 6 |
| *C. cuning* | Respirometry / mortality | 8 | 8 | 8 | 7 | 8 | 8 | 7 | 8 |

## Temperature acclimation

All fishes were either assigned to a control group (29.0 ± 0.5˚C; target temperature treatment ± range) or a high-temperature group (32.0 ± 0.5˚C). The control temperature replicated the average summer temperature experienced in the northern Great Barrier Reef (*Hughes et al., 2017*). The high-temperature treatment was chosen to simulate the estimated +3.0˚C already experienced during acute heating events and also projected as the mean rise in summer temperature by 2100 (*Collins et al., 2013*; *Garrabou et al., 2009*; *Hughes et al., 2017*; *IPCC, 2013*; *Pörtner et al., 2019*). In order to reach the experimental temperature for this treatment, water temperature was increased at a rate of 1.0˚C day$^{-1}$ until the target temperature was reached. Water temperature was maintained using automated heaters (5000W, Control Distributions, Carlton, Australia). To estimate the time required for acclimation, +3.0˚C exposed fishes were tested and sampled weekly for 4–5 consecutive weeks and compared to concurrent control samples taken in the beginning and end of the experimental period. To ensure a post-absorptive state, food was withheld from all fishes for 24 hr prior to experimentation or sampling (*Niimi and Beamish, 1974*).

## Respirometry

Oxygen uptake rates ($M$O$_2$) were obtained using intermittent-flow respirometry and used as a proxy for whole-animal aerobic metabolic rate (*Svendsen et al., 2016*). Three estimates of metabolic rate were examined: (1) SMR (metabolic rate of a fasted and resting individual), (2) MMR (the maximum aerobic metabolic rate that an individual can achieve, *Svendsen et al., 2016*), and (3) ASc (the scope for activities beyond basic maintenance) calculated as the absolute difference between MMR and SMR (*Chabot et al., 2016*; *Norin and Clark, 2016*; *Roche et al., 2013*). To measure $M$O$_2$, individual fish were placed into a 0.5 m (diameter) round aquarium containing well-aerated and temperature-controlled seawater (0.2 m deep), which was maintained at the same temperature as the fish's acclimation temperature (5000W, Control Distributions, Carlton, Australia). Fish were first chased continuously by hand for 3 min and then scooped into a mesh net and maintained out of the water for either 1 min (*C. cuning*) or 30 s (*C. quinquelineatus*) (*Clark et al., 2013*; *Killen et al., 2017*; *Roche et al., 2013*; *Rummer et al., 2016*). Following air exposure, fish were immediately placed into an acrylic cylindrical, intermittent-flow respirometry chamber (110 mL for *C. quinquelineatus* and 370 mL for *C. cuning* following the design of *Svendsen et al., 2016*) and submerged in an 800 L temperature-controlled water bath (5000W heater, Control Distributions, Carlton, Australia). Within 10 s of placing the fish into the chamber, post-exercise $M$O$_2$ was monitored continuously. This continued for a period of 7 min (*C. cuning*) or 9 min (*C. quinquelineatus*), after which time a submersible flush pump (Aquapro AP200LV, 200 L h$^{-1}$) flushed each chamber with aerated, temperature-controlled seawater from the surrounding water bath (3 min for both species). Following this, fish remained in chambers (with intermittent flushing cycles) to recover from exercise and reach SMR for an additional 18–24 hr, a time period deemed more than sufficient in similar reef fish species in past studies (*Chabot et al., 2016*; *Roche et al., 2013*). Each chamber was mixed continuously throughout each trial using a recirculating pump connected to the chamber in a closed loop, in order to ensure a homogenous concentration of oxygen throughout the chamber (*Svendsen et al., 2016*). A total of eight respirometry chambers were run in parallel. The 7- and 9-min measuring periods were chosen to ensure that $M$O$_2$ produced a linear decline in oxygen saturation within the chambers sufficient to calculate slope, but short enough to maintain oxygen saturation levels above 80% for each species (following *Svendsen et al., 2016*).

Throughout the entire respirometry trial, dissolved oxygen concentration (mg L$^{-1}$) of the water in each chamber was measured continuously (frequency 0.5 Hz) using an oxygen-sensing optode mounted in the recirculation loop, to ensure that flow was sufficient for a rapid response time of the sensor (*Svendsen et al., 2016*). These optodes were linked to two 4-channel Firesting Optical Oxygen Meters (Pyro Science, Aachen, Germany), which were connected to a PC that logged all oxygen data.

Raw text files were imported into LabChart v.6.1.3 (ADInstruments, Dunedin, New Zealand) and $M$O$_2$ (mg O$_2$ kg$^{-1}$ h$^{-1}$) was calculated from each slope (linear regression of oxygen concentration decline over time) throughout the trial using equations modified from *Bushnell et al., 1994* and *Schurmann and Steffensen, 1997*, taking into consideration the volume of the respirometry chamber, mass of the fish, and water displacement of the fish. The maximal oxygen uptake rate was

calculated during the first measurement period following exercise, defined as the steepest 1 min slope section with an $r^2$ >0.99, and used to estimate MMR. The SMR was estimated from the average of the lowest 10% of $MO_2$ values recorded during the 18–24 hr recovery period (*Clark et al., 2013*; *Rummer et al., 2014*; *Rummer et al., 2013*). Background $MO_2$ (calculated as an exponential increase based on measures of oxygen depletion in the empty respirometer before and after each trial) was subtracted from all $MO_2$ measurements. An exponential increase in background respiration was used to account for the typical growth rate of bacteria. To limit background respiration rates to less than 5% of a fish's SMR, all water flowed through a 36W UV filter (Blagdon Pro, UVC 16200, China) during experiments. Additionally, between all experiments, chambers and pumps were rinsed with a 10% bleach solution and fresh water and allowed to dry before commencing further trials.

## Blood and tissue analyses

Fishes were captured from holding tanks and isolated in flow-through holding containers within their respective holding tanks for 4 hr prior to blood and tissue sampling, as this minimized capture stress immediately prior to sampling and allowed all fish to be sampled in the same state. Immediately prior to sampling, all fish were euthanized by cranial concussion. All animals were then weighed ($M_b$; body mass, in g) and measured (*L*; standard length, in mm), so that Fulton's K Condition Factor could be calculated using the equation:

$$K = 100 * \left( M_b / L^3 \right).$$

For *C. cuning* only, blood was drawn from the caudal vein and collected in 1 mL heparinized syringes. Whole blood hemoglobin concentration was determined using a HemoCue (Hb 201 System, Australia Pty Ltd.) with 10 μL of whole blood (reported as grams per 100 mL using a calibration curve according to *Clark et al., 2008* corrected for tropical reef species by *Rummer et al., 2013*). Whole blood glucose and lactate concentrations (mM) were measured using 15 μl of blood in an Accutrend Plus (Roche Diagnostics Australia Pty Ltd.). Hamatocrit (Hct) was determined by centrifuging 60 μl of whole blood in heparinized micro-capillary tubes for 3 min and calculated as the ratio of packed RBCs to total blood volume (%). Both whole blood hemoglobin and Hct were used to calculate the mean corpuscular hemoglobin content (MCHC). Plasma volume was generally insufficient for other analyses (e.g. $Cl^-$, $HCO_3^-$, total $CO_2$, and catecholamines). No blood analyses were conducted on *C. quinquelineatus*, as it was not possible to draw a sufficient volume of blood. However, some blood oxygen transport properties were assessed by analyzing the spleen (e.g. spleen somatic index and spleen hemoglobin concentration).

For all animals, pectoral muscle and spleen tissue was dissected, flash frozen in liquid nitrogen, and stored at −80°C until later analyses. The second gill arch from each side of the fish was also dissected. One gill arch was flash frozen in liquid nitrogen and stored at −80°C for subsequent enzyme analyses, while the second gill arch was preserved in 10% neutral buffered formalin (replaced after 48 hr with 70% EtOH) for histological analyses. The spleen-somatic index (SSI) was calculated as the ratio of the spleen to body mass (expressed as a %). The spleen was then homogenized in 1 mL Drabkin's solution (Sigma-Aldrich cat. no. D5941; St. Louis, MO, USA) and absorbance was measured spectrophotometrically at 540 nm. Spleen hemoglobin concentration was then calculated by applying a millimolar extinction coefficient of 11 and accounting for spleen volume (by mass).

Frozen gill and red muscle tissues were homogenized in an enzyme extraction buffer (5 mM EDTA, 0.1% Triton X-100, 0.2 mM dithiothreitol, 50 mM HEPES (pH 7.4 at 25.0°C)) using a FastPrep−24 homogenizer (MP Biomedicals, Santa Ana, USA). The supernatant was stored at −80°C and used for the subsequent enzyme assays. All assays were performed on a temperature-controlled Spectramax Plus 384 Microplate Reader spectrophotometer (Molecular Devices, Sunnyvale, CA) at 25.0°C. Protein content (mg mL$^{-1}$) was determined using the Bradford's method with a standard curve built using serial dilutions of commercially available bovine serum albumin standards (BSA); this is an accepted and standard means for determining the total protein concentration of a homogenate (*Bradford, 1976*). LDH, an indicator of anaerobic energy production, was measured using a modified assay from *Johnston et al., 1977*. The assay buffer consisted of 50 mM Tris-HCl, 2 mM sodium pyruvate and 0.15 mM NADH (pH 8.0). Absorbance was measured at 340 nm for 3 min. CS, the first enzyme in the tricarboxylic acid (TCA) cycle and marker of mitochondrial density, was measured according to *McClelland, 2005*. The assay buffer contained 20 mM TRIS (pH 8.0), 0.1 mM

DTNB, and 0.3 mM acetyl-CoA. The reaction was initiated by the addition of 0.5 mM oxaloacetate, and absorbance was measured for 5 min at 412 nm. Control samples were assayed without oxaloacetate to control for background hydrolase activity.

Gill samples for histology were processed, embedded in paraffin wax, and sectioned at 5 μm. Four sections for each individual were taken, stained with hematoxylin and eosin (H and E), and examined under a light microscope (Leitz Wetzler, Germany). Digital micrographs of one section for each individual were taken using a Leica DFC310 FX microscope camera connected to Leica Application Suite v4.0.0 software (Leica Microsystems Limited, Switzerland) and processed using ImageJ v1.48 (Wayne Rasband, National Institutes of Health, USA). For each image, the gill arch was divided into three areas: ventral (bottom), medial (middle), and apex (top). In each area, 3– 5 filaments were chosen, but only if the filaments displayed good lamellar and filamental orientation, by exhibiting equal length lamellae on either side of the filament to ensure adequate embedding of the tissue. At three designated locations along the filament length (25%-near the base of the filament next to the gill arch, 50%- middle, and 75%-top), five lamellae were randomly chosen and traced. Their lamellar width, lamellar perimeter, and epithelial thickness (in mm) were measured using the ImageJ software (illustrated in *Figure 3*; *Bowden et al., 2014*). Perimeter was used as the primary proxy for surface area. In one instance (week 3 for *C. quinquelineatus*), there were insufficient samples with appropriate lamellar and filamental orientation (n = 2), and data for this week were therefore excluded from statistical analyses.

## Statistical analyses

All data were analyzed using general linear mixed effects (LME) models fit by restricted maximum likelihood (REML) following *Harrison et al., 2018*. When significant relationships were found, *post-hoc* planned comparisons (p.c.) on covariate-adjusted means were conducted to determine the differences between fish maintained at current-day summer average temperatures (control) to those maintained at an elevated temperature (+3.0℃) for 0, 1, 2, 3, 4, or 5 weeks. All p-values were corrected for type I error using False Detection Rate $p_{cutoff}$ (*Benjamini and Hochberg, 1995*).

In detail, four separate models were used to analyze the data: (1) metabolism (interactions between species, acclimation duration, and each parameter), (2) condition factor, biochemical indicators, and oxygen transport parameters for *C. cuning* (interactions between acclimation duration, and each parameter), (3) condition factor and biochemical parameters for *C. quinquelineatus* (interactions between acclimation duration, and each parameter), and (4) gill morphology (interactions between species, acclimation duration, and proxies for gill surface area). For metabolism (due to ethical requirements to minimize sample numbers), we used a mixed-staggered repeated measures design, in which each individual was tested twice where possible, but with a 3-week separation between each trial. This was done in order to prevent habituation to the chase protocol and thereby ensure consistent measures of maximal oxygen uptake. For blood, tissue, and gill, we compared the interaction between acclimation time and tissue for each species due to significant differences in the quantity and type of data obtained for each species. LMEs are highly robust to non-independence of data points obtained on the same individual and can produce unbiased estimates of variance and covariance (*Bolker et al., 2009*). In these models, we treated species and each parameter as fixed effects. Individual fishes were nested within species and treated as random effects to account for non-independence of repeated measures, while exposure period was nested within individual. Where possible, fish mass was treated as a continuous covariate and used to generate least squares mass-adjusted means for *post-hoc* comparisons. To assess the validity of the mixed effects analyses, we performed likelihood ratio tests comparing the models with fixed effects to the null models with only the random effects. We rejected results in which the model, including fixed effects, did not differ significantly from the null model. We checked for normality and homogeneity by visual inspections of plots of residuals against fitted values and used Coxbox transformations where appropriate (see statistical model routines and raw outputs in *Supplementary file 1* and model adjusted means in *Supplementary file 2*). Significance of main effects were estimated using upper- and lower-bound degrees of freedom and associated p-values (*Tremblay and Ransijn, 2015*) based on the Markov Chain Monte Carlo (MCMC) principle of pamer.fnc (*Bates and Maechler, 2009*). The use of upper- and lower-bound p-values is robust to the fact that the exact degrees of freedom cannot be calculated in complex LME designs (*Bates and Maechler, 2009*). Model strength was increased by pooling control data across exposure periods and numbers for each group are highlighted in associated

figure captions. All data were analyzed using the R (*R Development Core Team, 2016*) packages *lme4* (*Bates and Maechler, 2009*), *languageR* (*Baayen, 2013*), *LMERConvenienceFunctions* (*Tremblay and Ransijn, 2015*), *multcomp* (*Hothorn et al., 2008*), and *lsmeans* (*Lenth, 2016*). Planned comparisons are indicated in the text using subscripts (e.g. $p_{p.c.week3}$ denotes the planned comparison [p.] between control [c.] and week 3 [week3] individuals).

## Acknowledgements

We thank T Nay and L Holmes for assistance with fish husbandry and the staff at the Marine and Aquaculture Research Facilities Unit (MARFU) for logistical support. We also thank E Walsh for fish illustrations used in the figures. This research has been supported by funding from an Australian Research Council (ARC) Super Science Fellowship and Early Career Discovery Award to JLR, infrastructure and research allocation from the ARC Centre of Excellence for Coral Reef Studies at James Cook University to JLR, and by an Australian Postgraduate Award and International Postgraduate Research Scholarship to LEN.

## Additional information

### Funding

| Funder | Grant reference number | Author |
| --- | --- | --- |
| Australian Research Council | Super Science Fellowship | Jodie Rummer |
| Australian Research Council | Early Career Discovery Award | Jodie Rummer |
| James Cook University | Infrastructure and research allocation | Jodie Rummer |
| James Cook University | Australian Postgraduate Award | Lauren E Nadler |
| James Cook University | International PostgraduateResearch Scholarship | Lauren E Nadler |

The funders had no role in study design, data collection and interpretation, or the decision to submit the work for publication.

### Author contributions

Jacob L Johansen, Conceptualization, Data curation, Software, Formal analysis, Supervision, Validation, Investigation, Methodology, Writing - original draft, Project administration, Writing - review and editing; Lauren E Nadler, Conceptualization, Data curation, Validation, Investigation, Methodology, Writing - original draft, Project administration, Writing - review and editing; Adam Habary, Investigation, Writing - review and editing; Alyssa J Bowden, Data curation, Formal analysis, Methodology, Writing - review and editing; Jodie Rummer, Conceptualization, Resources, Data curation, Supervision, Funding acquisition, Investigation, Visualization, Methodology, Project administration, Writing - review and editing

### Author ORCIDs

Jacob L Johansen (ID) https://orcid.org/0000-0002-2912-2146
Lauren E Nadler (ID) https://orcid.org/0000-0001-8225-8344
Adam Habary (ID) https://orcid.org/0000-0002-1184-5581
Alyssa J Bowden (ID) https://orcid.org/0000-0001-6024-1891
Jodie Rummer (ID) https://orcid.org/0000-0001-6067-5892

### Decision letter and Author response

Decision letter https://doi.org/10.7554/eLife.59162.sa1
Author response https://doi.org/10.7554/eLife.59162.sa2

## Additional files

### Supplementary files
- Supplementary file 1. Statistical model routines and raw outputs.
- Supplementary file 2. Model adjusted means.
- Transparent reporting form

### Data availability
All metadata are available in the supplementary files.

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

## Appendix 1

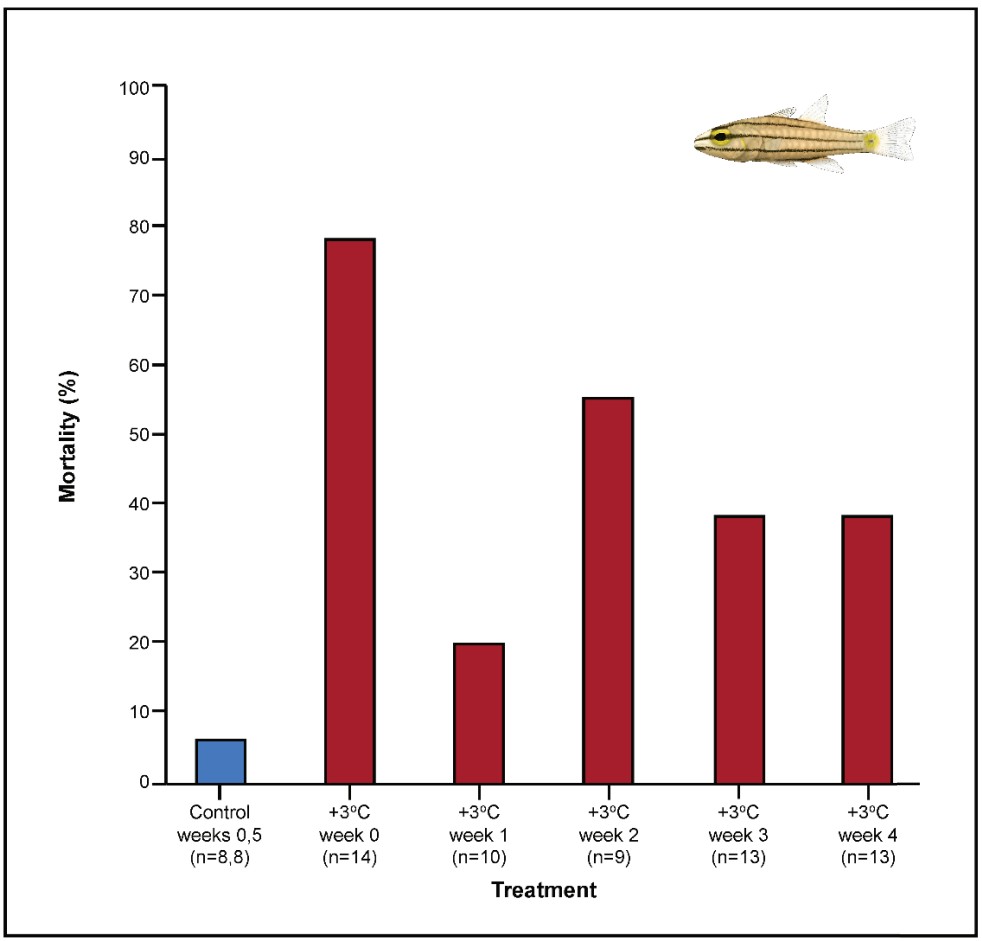

**Appendix 1—figure 1.** Effect of exhaustive exercise on the mortality (%) of *Cheilopterus quinqueli-neatus*. The first blue column illustrates control mortality (29.0°C), while red columns illustrate elevated temperature effects (+3.0°C). No mortality was seen in *Caesio cuning* during the experimental period (not depicted). Numbers in parentheses below each category on the x-axis denote sample sizes for each group.

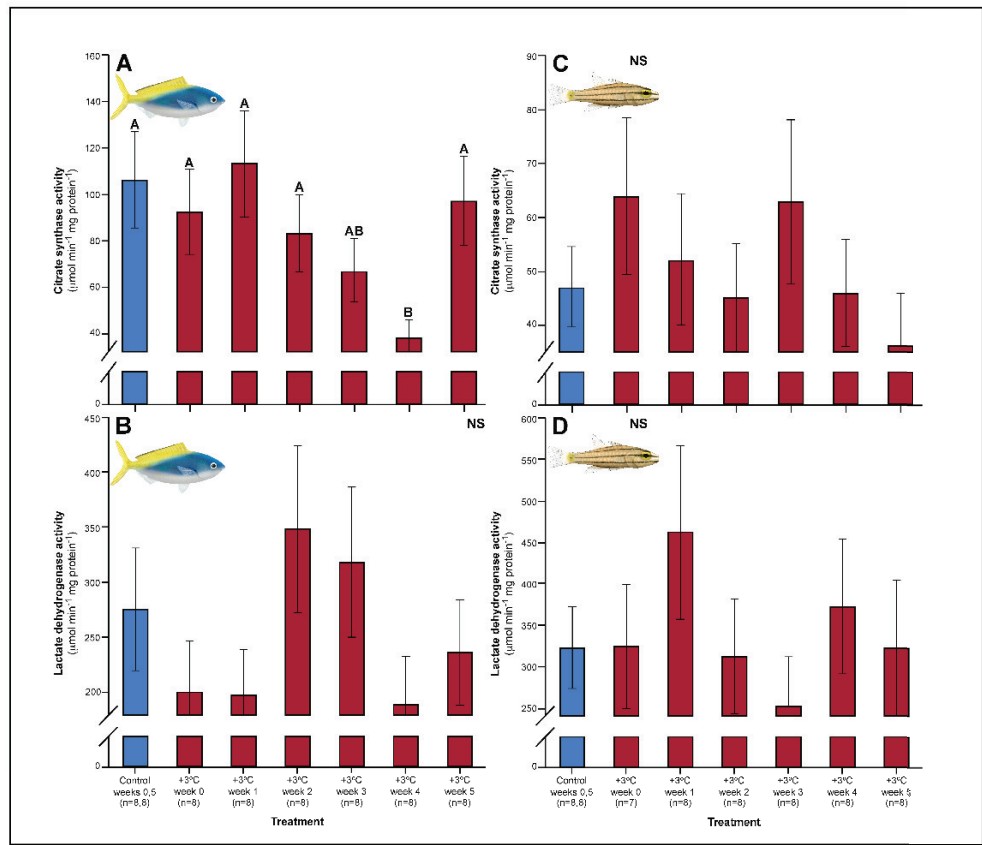

**Appendix 1—figure 2.** Effect of 5 weeks' exposure to elevated temperatures (+3.0°C) on citrate syn-thase (CS) and lactate dehydrogenase (LDH) activity in the gills of *Caesio cuning* (A and B) and *Chei-lopterus quinquelineatus* (C and D). The first blue column in each figure illustrates the control (29.0°C). Letters above columns indicate significant differences between treatment groups, determined through multiple comparisons *post-hoc* testing (based on linear mixed-effects model analysis). 'NS' denotes that the model did not indicate any significant effects of temperature. Error bars are s.e.m., and numbers in parentheses below each category on the x-axis denote sample sizes for each group.

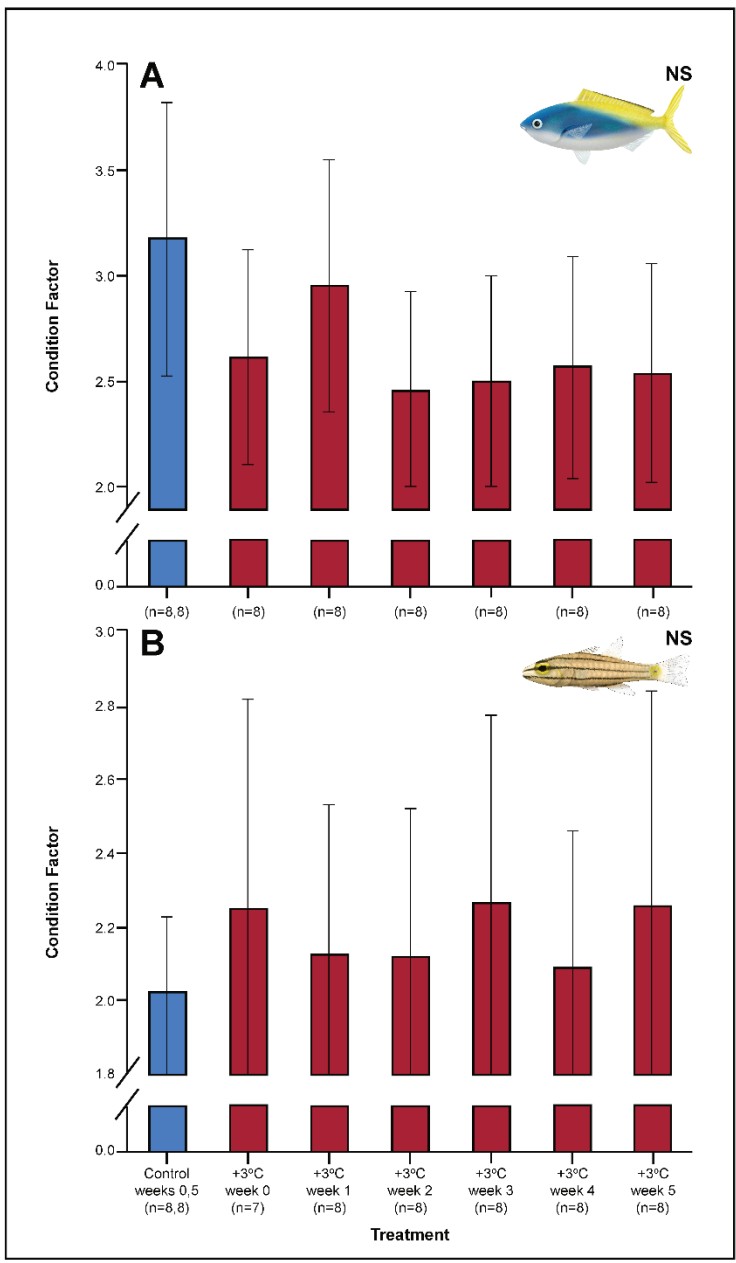

**Appendix 1—figure 3.** Effect of 5 weeks' exposure to elevated temperatures (+3.0°C) on Fulton's K condition factor in *Caesio cuning* (**A**) and *Cheilopterus quinquelineatus* (**B**). The first blue column in each figure illustrates the control (29.0°C). 'NS' denotes that the linear mixed-effects model analysis did not indicate any significant effects of temperature. Error bars are s.e.m., and numbers in parentheses below each category on the x-axis denote sample sizes for each group.

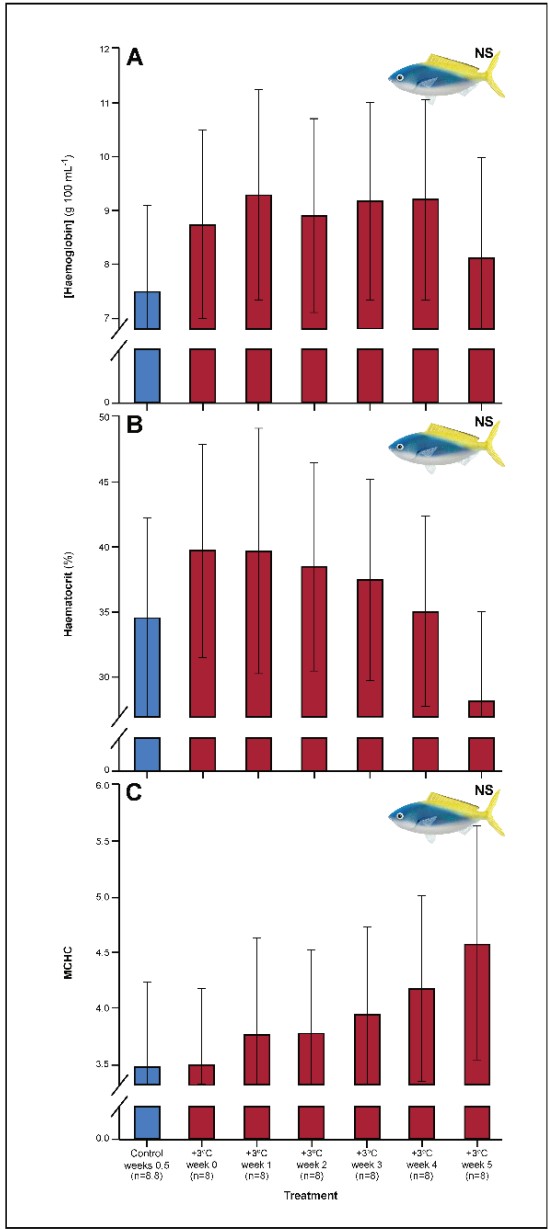

**Appendix 1—figure 4.** Effect of 5 weeks' exposure to elevated temperatures (+3.0℃) on (**A**) hemoglobin concentration, (**B**) hematocrit, and (**C**) mean corpuscular haemoglobin content (MCHC) in *Caesio cuning*. The first blue column in each figure illustrates the control (29.0℃). 'NS' denotes that the linear mixed-effects model analysis did not indicate any significant effects of temperature. Error bars are s.e.m., and numbers in parentheses below each category on the x-axis denote sample sizes for each group.

## Appendix 2

This appendix describes parameter-specific changes over time according to the level of organization, starting from the organism to the cellular level and grouped according to energy/oxygen demand and supply.

### Mortality

Among the +3.0°C-exposed individuals, 47.5% of all *C. quinquelineatus* died over the 5-week period. By comparison, fewer than 6% of control *C. quinquelineatus* died over the same period. The highest mortality rates were recorded after exhaustive exercise among the +3.0°C-exposed individuals examined for metabolic performance, as individuals of *C. quinquelineatus* showed a 79, 20, and 56% mortality in weeks 0, 1, and 2, respectively, and a 38.5% mortality rate during weeks 3 and 4 (*Appendix 1—figure 1*). No mortality was recorded for *C. cuning* during these trials. Both species maintained Fulton's K condition factor (i.e. body condition, *Appendix 1—figure 3*) across all treatments, suggesting that the mortality rates detected in +3.0°C-exposed individuals were due to changing temperature conditions rather than waning health under laboratory confinement.

### Whole organism demand

Within 1 week of exposure to the elevated temperature treatment, *C. cuning* exhibited changes in metabolic oxygen demand expressed as a higher SMR ($p_{p.c.week0}$ = 0.021), which returned to control levels after 1 week of acclimation (*Figure 5*). In comparison, *C. quinquelineatus* only showed a moderate but non-significant increase in SMR ($p_{p.c.week0} > P_{cutoff}$) over the same period and a brief reduction of SMR below control values in week 3 (SMR: $p_{p.c.week3}$ = 0.002) before returning to control values from week 4 onwards (*Figure 5*).

### Whole organism supply

Within the first week of elevated temperature exposure, *C. cuning* exhibited moderate but non-significant increases in MMR and ASc, culminating in significant peaks after 3 weeks' exposure to elevated temperatures when compared to controls (MMR, $p_{p.c.week3}$ = 0.013; ASc, $p_{p.c.week3}$ = 0.017). Both MMR and ASc returned to control levels by week 4 (*Figure 5*). Conversely, elevated temperature exposure caused an immediate increase in MMR and ASc in *C. quinquelineatus* (MMR: $p_{p.c.week0}$ <0.001; ASc: $p_{p.c.week0}$ <0.001) lasting past the second week of exposure (MMR: $p_{p.c.week2}$ = 0.002; ASc: $p_{p.c.week2}$ <0.001). In this species, MMR exhibited a brief reduction below control values in week 3 ($p_{p.c.week3}$ = 0.015) before returning to control values by week 4. ASc returned to control levels by week 3 (*Figure 5*).

### Tissue demand

In the gills and pectoral muscle of *C. cuning*, CS activity declined gradually to a significant minimum in week 4 (Gill: $p_{p.c.week4}$ <0.001, Muscle: $p_{p.c.week4}$ = 0.011) before returning towards control levels in week 5 ($p_{p.c.week5}$ >0.5; *Figure 2*, *Appendix 1—figure 2*). Conversely, *C. quinquelineatus* showed no change in gill CS activity, but pectoral muscle CS activity reached a peak in week 4, before returning to control levels by week 5 ($p_{p.c.week4}$ = 0.004; *Figure 2*, *Appendix 1—figure 2*).

While neither species exhibited significant gill plasticity in the anaerobic metabolic enzyme, LDH over the 5-week exposure period (*Appendix 1—figure 2*), within the muscles, *C. cuning* showed an immediate decline in LDH from 1.36 to 0.61 mM compared to controls, ($p_{p.c.week0}$ = 0.003; *Figure 2*). In *C. quinquelineatus*, muscle LDH was consistent with control levels from weeks 0 to 2 before showing a significant peak in week 3 ($p_{p.c.week3}$ = 0.021) and a significant depression in LDH activity in week 5 ($p_{p.c.week5}$ = 0.016; *Figure 2*).

### Tissue supply

In the gills, lamellar width increased in *C. cuning* ($p_{p.c.week1}$ = 0.002) within one week of exposure to the elevated temperature treatment, a trend that continued through weeks 2 and 3 ($p_{p.c.week2}$ <0.001; $p_{p.c.week3}$ <0.001; *Figures 3* and *4*). This was augmented in week 3 by a significant

increase in total diffusible lamellar perimeter ($p_{p.c.week3}$ <0.001; *Figure 3*). In comparison, *C. quinquelineatus* showed no acute changes to gill structure until week 2, when total lamellar perimeter decreased significantly ($p_{p.c.week2}$ <0.001; *Figure 3*). Neither species showed a significant response in epithelial thickness. All gill morphology parameters had returned to control values by week 4 in both species ($p_{p.c.week4}$ > $P_{cutoff}$).

In the blood, *C. cuning* showed changes relative to control within the first week of exposure to elevated temperatures. Blood glucose levels more than doubled from weeks 0 to 1, from 1.93 to 4.65 mM ($p_{p.c.week1}$ = 0.003) and remained elevated through week 2 ($p_{p.c.week2}$ = 0.029), returning to control levels from week 3 onwards (*Figure 1*). In comparison to control fish, this species showed no change in blood [lactate] (*Figure 1*), blood hematocrit, whole blood [Hb], mean corpuscular hemoglobin content [MCHC] (*Appendix 1—figure 4*), or spleen somatic index ($p_{p.c.week5}$ > $p_{cutoff}$; *Figure 6*, *Appendix 1—figure 2*), providing no indication of changing RBC circulation. However, spleen [Hb] exhibited a non-significant trend to increase over the entire 5-week period culminating in a significant peak in week 5 relative to week 0 ($p_{p.week0.week5}$=0.007), indicative of ongoing compensatory mechanisms for blood oxygen transport (*Figure 6*).

*C. quinquelineatus* could not be tested for blood parameters and showed no significant change in spleen [Hb] over the 5-week exposure period. It did, however, exhibit a significant 56.2% reduction in spleen somatic index ($p_{p.c.week5}$ <0.001, *Figure 6*) in week 3, which continued beyond the end of the 5-week exposure period.

