## [Decision Letter]

**Acceptance summary:**

Your study provides valuable insight into the time courses of metabolic and physiological changes that occur in response to heat waves in fishes that can only tolerate small temperature changes. This will hopefully allow for better predictions of the effects of climate change on coral reef ecology. It also provides information on required study durations to capture the acclimation and performance capacity of fishes (and likely other animals) in future experiments.

**Decision letter after peer review:**

Thank you for submitting your article "Thermal acclimation of tropical coral reef fishes to global heat waves" for consideration by *eLife*. Your article has been reviewed by three peer reviewers (one review was a joint review), and the evaluation has been overseen by a Reviewing Editor and Christian Rutz as the Senior Editor. The following individual involved in the review of your submission has agreed to reveal their identity: Simon Morley (Reviewer #2).

The reviewers have discussed their reviews with one another, and the Reviewing Editor has drafted this decision letter to help you prepare a revised submission.

We would like to draw your attention to changes in our revision policy that we have made in response to COVID-19 (https://elifesciences.org/articles/57162). Specifically, we are asking editors to accept without delay manuscripts, like yours, that they judge can stand as *eLife* papers without additional data, even if they feel that they would make the manuscript stronger. Thus the revisions requested below are important, but only address clarity and presentation.

All reviewers agree that the study will provide important data for a better understanding how stenothermal species respond to heat waves. However, there are several major points that require attention. We believe that all points can be addressed by carefully amending the text.

Essential revisions:

1) The experimental approach studying the physiological responses of two warm-stenothermal fish species with different ecological background and life styles is highly valuable. However, this approach is missing in the authors' hypotheses. The sentence given ("may alter their physiological response…") is too vague. The authors should be more specific in their prediction. In what way do the authors expect the physiological responses to differ between the two species? Considering the species comparison, the hypothesis given ("that it would take a minimum of three weeks…") is confusing, and seems to ignore the large amount of interesting background data available.

2) Considering the aim of the study, and the existing literature on thermal responses to warming, it is astonishing that the Introduction lacks physiological concepts such as GOLD (see literature by D. Pauly for gill morphology) and OCLTT (see H.O. Pörtner for oxygen supply and demand and aerobic capacity). These concepts need to be incorporated. This will also allow the authors to tell a stronger, more coherent story (Introduction).

3) The Discussion would benefit considerably from integrating the observed physiological changes with the above-mentioned concepts. Furthermore, a more in-depth discussion of the thermal response of cold-stenothermal fish is recommended (e.g., Lucassen et al. AJP 2004 on thermal response of the citrate synthase; Windisch et al. MolEcol 2014 for a metabolic shift to carbohydrate-based metabolism; and Rebelein et al. CBP 2018 on the thermal response of gill tissue of two related cold-stenothermal fishes).

4) We also advise to restructure the Results section. In the present form, it is hard to follow parameter-specific changes over time. We recommend to present the responses according to the level of organization, starting from the organism to the cellular level. We also suggest to group parameters belonging to energy/oxygen supply and demand, respectively. This includes Table 1. This will make it much easier to follow the main findings in this extensive dataset.

5) The exact number of biological replicates needs to be added to each figure in order to allow a transparent and fast assessment of the strength of the presented datasets.

---

## [Author Response]

Essential revisions:1) The experimental approach studying the physiological responses of two warm-stenothermal fish species with different ecological background and life styles is highly valuable. However, this approach is missing in the authors' hypotheses. The sentence given ("may alter their physiological response…") is too vague. The authors should be more specific in their prediction. In what way do the authors expect the physiological responses to differ between the two species? Considering the species comparison, the hypothesis given ("that it would take a minimum of three weeks…") is confusing, and seems to ignore the large amount of interesting background data available.

Thank you for this comment. We agree and have now included more detail on our hypotheses and expectations as requested:

“These species are also separated by more than 100 million years of evolution (Near et al., 2013) and differ in most major life-history characteristics including lifespan (<2 versus >8 years), habitat use (site-attached versus roaming), and mobility (sedentary versus mobile) for *C. quinquelineatus* and *C. cuning* respectively (Randall, Allen and Steene, 1997). We hypothesised that these differences in life history characteristics may alter physiological responses to unfavourable thermal conditions, with *C. quinquelineatus* less responsive to short-term perturbations due to its sedentary and site-attached life-history while *C. cuning* would execute immediate physiological changes when unable to use its mobility to outmanoeuvre unfavourable conditions”.

This is further supported by:

“Based on previous reports of the timing and duration of physiological responses of stenothermic fishes to elevated temperatures (e.g., Madeira et al., 2016; Sidell et al., 1973; Somero, 2015), we hypothesized that it would take a minimum of three weeks for the putative “slower acclimating species” (*C. quinquelineatus*) to stabilize all hematological and cardiorespiratory parameter following elevated temperature exposure”.

2) Considering the aim of the study, and the existing literature on thermal responses to warming, it is astonishing that the Introduction lacks physiological concepts such as GOLD (see literature by D. Pauly for gill morphology) and OCLTT (see H.O. Pörtner for oxygen supply and demand and aerobic capacity). These concepts need to be incorporated. This will also allow the authors to tell a stronger, more coherent story (Introduction).

Thank you. We have now rewritten the Introduction to include an overview of the hotly debated theoretical expectations for ectothermic responses under elevated temperatures, including GOLT and OCLTT. It now reads:

“The capacity for most species to maintain fitness under the rapid incursion of anthropogenic climate change remains uncertain. […] Recent reviews have therefore emphasized the urgent need for cross-disciplinary, mechanistic studies that explore the timescales over which thermal responses occur as well as studies that assess the processes associated with acclimation and adaptation in thermally sensitive species (Audzijonyte et al., 2019; Jutfelt et al., 2018)”.

3) The Discussion would benefit considerably from integrating the observed physiological changes with the above-mentioned concepts. Furthermore, a more in-depth discussion of the thermal response of cold-stenothermal fish is recommended (e.g., Lucassen et al. AJP 2004 on thermal response of the citrate synthase; Windisch et al. MolEcol 2014 for a metabolic shift to carbohydrate-based metabolism; and Rebelein et al. CBP 2018 on the thermal response of gill tissue of two related cold-stenothermal fishes).

We concur that the Discussion would benefit from integration with the GOLT and OCLTT theories, particularly in terms of how these data conform (or do not) to the theoretical predictions stemming from each theory. We have now included a paragraph on this topic in the Discussion, which now reads:

“The devastating impacts of climate change on marine resources reinforces the urgent need for cross-disciplinary studies about the timescales, mechanisms, and limitations of thermal responses in ectothermic marine species (Audzijonyte et al., 2019; Jutfelt et al., 2018). […] Our results do, however, support the idea that cardio-respiratory transport and tissue demand are primary determinants of an organism’s performance under ocean warming (see extended discussions in Ern, 2019; H. O. Pörtner, 2014; Sandblom et al., 2016), as an increase in total energy supply appeared to support *C. cuning* through the five-week acclimation period; whereas a lack of aerobic energy availability was associated with mass mortality in *C. quinquelineatus*”.

Please note, the reviewers asked us to remove the section about thermal tolerance of different life stages (see comments below), as it is not the topic of the present study. We concur that the discussions should stay on topic and have removed the section as requested. Similarly, this is a study of thermal responses in warm-adapted tropical stenothermal fishes, and the request for an in-depth discussion of cold-stenothermic fishes therefore seem off topic as well. We have opted to instead include several references to cold adaption throughout the manuscript, rather than adding a dedicated section to comprehensively discuss cold adaptation (see e.g. Introduction, subsection “Useful biomarkers of thermal stress and ongoing acclimation”).

4) We also advise to restructure the Results section. In the present form, it is hard to follow parameter-specific changes over time. We recommend to present the responses according to the level of organization, starting from the organism to the cellular level. We also suggest to group parameters belonging to energy/oxygen supply and demand, respectively. This includes Table 1. This will make it much easier to follow the main findings in this extensive dataset.

Thank you for this suggestion. We already tried the suggested structuring of the Results (including providing the description in accordance with level of organization, by oxygen supply and demand, and from cellular to whole organisms). After careful consideration to the tradeoffs of each approach, we finally settled on a description that follows the sequence of events over time. We recognize that cellular physiologists and cardio-respiratory experts are likely to want the results presented at the specific organizational level of their own expertise (i.e., parameter-specific), and members of our team are no exception. However, we also concluded that readers looking for broader patterns that transcend specific enzymes, cells, tissues, or organs are likely to look at the time-series of events. Similarly, ecologists aiming to understand fitness-related consequences for survival in the wild are likely to look at the data as a whole, and across discrete time intervals. We recognize that our largest audience will likely be ecologists and eco-physiologists working with whole organism responses to climate change, such as scientists working with larger unifying theories beyond GOLT and OCLTT or needing to acclimate fishes in the lab before testing large scale predictions. For instance, many studies presently only acclimate fishes for one or two weeks before conducting climate change related experiments, but our results suggest that this is inadequate. We have taken all of these differing needs into consideration, and firmly believe the current structure and delivery of results provide the most parsimonious overview with the capacity to satisfy most readers to the greatest extent. Each parameter’s relationship with energy/oxygen supply versus demand are clearly delineated in the text (e.g., see: “blood glucose and muscle LDH activity as indicators of anaerobic energy production (Nishad Jayasundara et al., 2013; Windisch et al., 2011); splenic RBC stores (SSI and Spleen [Hb]) denoting altered oxygen transport capacity (Ken-Ichi, 1988); and lamellar perimeter and width, illustrating changes to oxygen uptake capacity over the gills (Sollid and Nilsson, 2006)”).

We want to ensure that our study is maximally useable for all, and as such, we do see merit in parameter-specific presentation. Therefore, we have now included an additional Appendix that provides a clear delineation of parameter-specific responses, starting with enzymes and ending with whole body metabolism (see Appendix 2). We have also included a new figure (Figure 7) which visualizes the chronology of parameter specific responses, aimed at making our results easier to follow and reference.

5) The exact number of biological replicates needs to be added to each figure in order to allow a transparent and fast assessment of the strength of the presented datasets.

Thank you for this suggestion. We have now included n-values for all replicated in our figures.